# CORE: PERFORMANCE-LOSSLESS CONTEXT COMPRESSION FOR RETRIEVAL-AUGMENTED GENERATION

## ABSTRACT

Retrieval-Augmented Generation (RAG) has emerged as a promising approach to enhance the timeliness of knowledge updates and the factual accuracy of responses in large language models. However, incorporating a large number of retrieved documents significantly increases input length, leading to higher computational costs. Existing approaches to document compression tailored for RAG often degrade task performance, as they typically rely on predefined heuristics in the absence of clear compression guidelines. These heuristics fail to ensure that the compressed content effectively supports downstream tasks. To address these limitations, we propose CORE, a novel method for lossless context compression in RAG. CORE is optimized end-to-end and does not depend on predefined compression labels, which are often impractical to obtain. Instead, it leverages downstream task performance as a feedback signal, iteratively refining the compression policy to enhance task effectiveness. Extensive experiments across four datasets demonstrate the effectiveness of CORE. With a high compression ratio of 3%, CORE not only prevents performance degradation compared to including full documents (i.e., without compression) but also improves the average Exact Match (EM) score by 3.3 points. The code for CORE is available at `https://anonymous.4open.science/r/CORE-28B4`.

## 1 INTRODUCTION

Large language models (LLMs) have undergone rapid development in recent years, significantly enhancing performance across various language tasks due to their emergent capabilities in semantic understanding and reasoning. Nevertheless, LLMs still face challenges in updating knowledge and providing factual responses (Fan et al., 2024). To address these issues, Retrieval-Augmented Generation (RAG) has emerged as a promising approach. By retrieving the most relevant documents from external knowledge bases and prepending them as contextual information to the original input, RAG substantially improves LLM performance on knowledge-intensive tasks (Ram et al., 2023).

While RAG enhances performance, its effectiveness is closely tied to the number of retrieved documents used, since a broader context increases the probability of encompassing critical evidence. As illustrated in Figure 1, performance was weakest without any retrieved documents (i.e., without RAG). Accuracy improved consistently as more documents were added to the context, ultimately exceeding the no-RAG baseline by over 10 Exact Match (EM) points. However, this performance gain came with two significant limitations: (1) a substantial increase in computational cost from processing a larger number of context tokens (Xu et al., 2024), and (2) the model's difficulty in effectively leveraging all provided documents, often resulting in the omission of key information located in the middle of the context (Liu et al., 2023).

These limitations have motivated recent research efforts aimed at compressing the retrieved context (Jin et al., 2024b; Wu et al., 2025; Jin et al., 2024a; Zhang et al., 2024a). Prominent approaches include document summarization (Xu et al., 2024), key information extraction (Cao et al., 2024; Xu et al., 2024), the construction of key supporting evidence (Jin et al., 2024b), and noise filtering based on information theory (Zhu et al., 2024). Despite recent progress, these methods have several notable shortcomings. First, compression often results in a performance trade-off. For instance, RECOMP

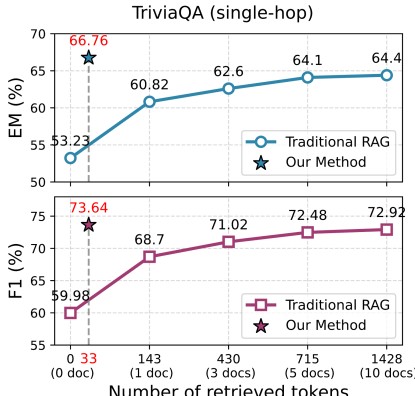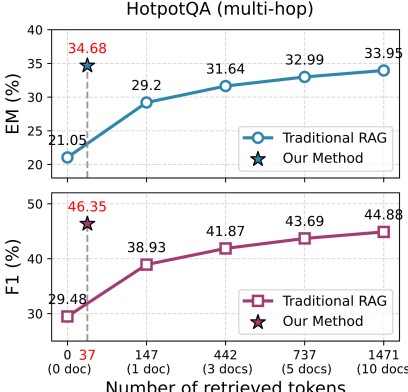

Figure 1: Performance evolution with an increasing number of retrieved documents on two datasets. Traditional RAG requires more documents for better performance, while our method achieves comparable or superior results with significant token compression.

(Xu et al., 2024) suffers a 3–5 point drop in EM score (Table 1), making it unsuitable for accuracy-sensitive applications. Second, most compression methods are heuristic in nature. The models are typically trained to generate summaries that are generally good but not necessarily useful for the downstream answer-generation LLM. This limitation arises from the lack of an ideal supervisory signal that defines what an optimal summary should be for the downstream task. This fundamental gap hinders the end-to-end optimization of existing methods. Finally, some compression models (Zhu et al., 2024) have parameter counts comparable to the LLM that performs the end task, resulting in substantial computational costs that undermine the efficiency gains of compression.

Addressing these critical shortcomings requires a method that aligns compression with downstream task requirements, thereby minimizing performance trade-offs. To bridge this gap, we propose **CORE**, a novel method designed to achieve lossless context compression for RAG. Unlike previous compression methods, CORE is optimized in an end-to-end manner. Since obtaining predefined summary labels for supervision is impractical, we instead use downstream task performance as a feedback signal to evaluate the compression model's output. This feedback enables iterative refinement of the compression policy, guiding the compression model toward improved downstream performance. To this end, we employ Group Relative Policy Optimization (GRPO), a technique particularly well-suited for this purpose (Liu et al., 2024; Shao et al., 2024; Chen et al., 2025a). In our framework, the accuracy of the downstream QA task is formalized as a reward, with the compression policy optimized through group-wise relative comparisons. Furthermore, our compression model is substantially smaller than the downstream LLM, which significantly reduces the computational overhead associated with encoding retrieved documents.

We evaluate CORE on four benchmark datasets: two single-hop QA datasets (*Natural Questions* and *TriviaQA*) and two multi-hop datasets (*HotpotQA* and *2WikiMultihopQA*). As shown in Table 1, CORE achieves state-of-the-art performance across all baselines. With a compression ratio of 3%, our approach not only avoids performance degradation compared to prepending full documents but also improves the average EM score by 3.3 points. We further demonstrate two key advantages of our approach. First, the effectiveness of CORE is not tied to a specific model architecture, as demonstrated by the fact that lossless compression can be achieved when various models are trained as compressors (Figure 3). Second, the compressor exhibits strong transferability: a compression model trained using feedback from one LLM generalizes effectively to other LLMs (Table 2). Finally, an in-depth case study (Tables 4 and 5) provides a qualitative analysis of the benefits of CORE.

## 2 CORE-RAG

This section introduces our proposed method, **CO**mpression via **RE**inforcement learning (**CORE**), which is shown in Figure 2. First, we provide an overview of the entire workflow. Then, we detail the end-to-end training strategy for the compression model, which is designed to drastically reduce the number of document tokens while preserving task performance.

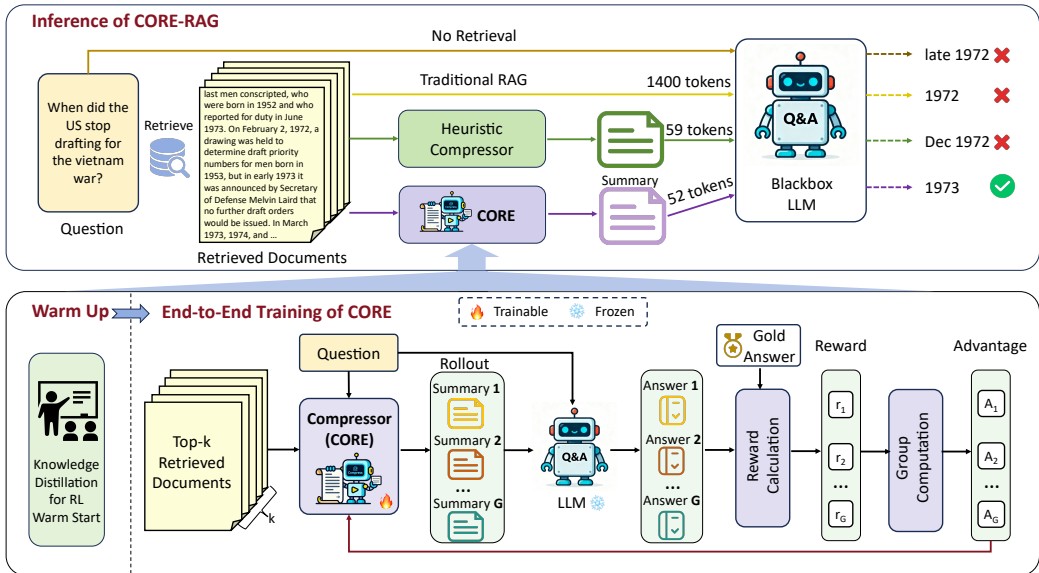

Figure 2: Overview of our method CORE. The upper section illustrates the inference pipeline. The lower section depicts the end-to-end training method for the compression model.

## 2.1 PROBLEM FORMULATION

We adopt the same problem formulation as prior work (Xu et al., 2024). Given an input question $q$, a target output $y$, and a set of $k$ retrieved documents $D$, our objective is to compress $D$ with respect to $q$ into a summary $s$ that preserves the most useful information while using significantly fewer tokens than $D$. This summary $s$ is then prepended to the original input $q$ and fed into an LLM to generate the final answer for the downstream task. This overall pipeline is illustrated in the upper half of Figure 2. The process involves two key components: a compression model $\pi_\theta \colon (q, D) \mapsto s$ and a large language model $M \colon (s, q) \mapsto \hat{y}$, which generates the predicted answer $\hat{y}$. We treat $M$ as a black-box system and focus exclusively on training the compressor $\pi_\theta$. The compressor itself is also a language model, but it is intentionally designed to be significantly smaller than $M$ to reduce the computational cost of encoding the retrieved documents.

## 2.2 TRAINING THE COMPRESSOR

Our compressor is designed to generate document summaries that are highly useful to the LLM ($M$) for downstream tasks. This objective is challenging because the criteria for an effective summary are task-dependent, and direct supervision is unavailable. We therefore formulate this as an end-to-end training problem and employ reinforcement learning to optimize the compressor without relying on pre-defined compression labels. The overall architecture of our training framework is illustrated in the lower portion of Figure 2. The following sections describe the key components of our approach: distillation warm-up, policy optimization, and reward calculation.

### 2.2.1 DISTILLATION FOR WARM-START

Due to the limited parameter size of our compression model, its capability for question-focused document summarization is constrained. To provide a strong initial policy for RL and ensure training stability, we employ knowledge distillation from a teacher model to initialize our compressor. Specifically, we first utilize a large-scale language model (DeepSeek-V3) as the teacher to generate summaries of retrieved documents related to the given question. We then evaluate the performance of the downstream LLM ($M$) on the QA task under two conditions: (1) with the teacher-generated summary $\hat{s}$ prepended to the input question $q$, and (2) with the original question alone. The corresponding performance scores are denoted as $p_{\text{summary}}$ and $p_{\text{original}}$, respectively. By comparing these results, we retain instances where $p_{\text{summary}} > p_{\text{original}}$, indicating that the summary enhances RAG performance. We also retain cases where $p_{\text{original}} = 1$ (i.e., the model produces a fully correct answer without the summary) and $p_{\text{summary}} < p_{\text{original}}$; for these, we set the target summary $\hat{s}$ to an empty string. All other instances are discarded. The resulting filtered and modified dataset is denoted as

$\mathcal{X}_f$, which is used for supervised fine-tuning of the compression model. The fine-tuning objective is defined as:

$$\mathcal{L}_{\text{distill}} = \frac{1}{|\mathcal{X}_f|} \sum_{(q,D,\hat{s}) \in \mathcal{X}_f} \mathcal{L}_{\text{CE}}(\pi_\theta(q, D), \hat{s}), \tag{1}$$

where $\pi_\theta(q, D)$ denotes the output of the compression model and $\mathcal{L}_{\text{CE}}$ is the cross-entropy loss. This distillation process yields a robust initialization for RL and promotes stability in subsequent training.

### 2.2.2 END-TO-END TRAINING WITH RL

Following the distillation phase, the compressor possesses a preliminary compression capability. However, as summaries from even the largest teacher models are not guaranteed to be optimal for the downstream task, further end-to-end optimization is necessary. We therefore formulate this optimization as a reinforcement learning problem. In this framework, the compressor functions as a policy that generates a summary from an input question and its corresponding documents. A reward function, which directly reflects performance on the downstream task (e.g., question-answering accuracy), then evaluates the summary's quality. The objective is to optimize the compressor's parameters to maximize the expected cumulative reward, thereby directly aligning its outputs with the downstream task's objectives.

Specifically, we employ Group Relative Policy Optimization (GRPO) (Shao et al., 2024) algorithm. Unlike Proximal Policy Optimization (PPO), which trains a separate critic model, GRPO estimates the baseline directly from a group of rollouts. Given an existing policy, $\pi_{\theta_{\text{old}}}$, and a reference policy, $\pi_{\theta_{\text{ref}}}$, the GRPO objective maximizes the compressor policy $\pi_\theta$ using $G$ rollouts $\tau = \{s_i\}_{i=1}^{G} \sim \pi_{\theta_{\text{old}}}(\cdot|x)$, for each input $x \sim \mathcal{D}$:

$$\mathcal{J}(\theta) = \mathbb{E}_{x \sim \mathcal{D},\ \{s_i\}_{i=1}^{G} \sim \pi_{\theta_{\text{old}}}(\cdot|x)} \tag{2}$$

$$\frac{1}{G} \sum_{i=1}^{G} \left[ \min\left( \frac{\pi_\theta(s_i|x)}{\pi_{\theta_{\text{old}}}(s_i|x)} A_i,\ \text{clip}\left( \frac{\pi_\theta(s_i|x)}{\pi_{\theta_{\text{old}}}(s_i|x)}, 1 - \epsilon, 1 + \epsilon \right) A_i \right) - \beta \mathbb{D}_{\text{KL}}\left( \pi_\theta \parallel \pi_{\theta_{\text{ref}}} \right) \right],$$

where $A_i = (r_i - \text{mean}(\{r_j\}_{j=1}^{G}))/\text{std}(\{r_j\}_{j=1}^{G})$ represents the normalized advantage of the $i$-th rollout within the group, $\epsilon$ is the clipping ratio, and $\beta$ is the coefficient for the KL divergence penalty. The inclusion of the KL divergence term ensures that the updated policy does not deviate significantly from the reference policy.

### 2.2.3 REWARD CALCULATION

**Generating End-Task Output.** It is important to note that the reward is not computed directly from the compressor's output summary. Instead, the summary $s$ generated by the compressor is prepended to the original input question $q$, and this combined input is fed into the LLM $M \colon (s, q) \mapsto \hat{y}$ to produce a predicted answer $\hat{y}$. The reward is then calculated by comparing $\hat{y}$ to the gold answer $y$. Throughout the training process, the parameters of $M$ remain fixed and are not updated.

**Computing Rewards.** We design simple rule-based rewards based on end-task performance to guide the compressor's improvement, which consists of two components:

- **EM Reward** ($r_{\text{EM}}$). We employ EM as the main reward function, which is a widely adopted metric for evaluating the accuracy of QA tasks. The EM reward is binary: it yields a value of 1 if the generated answer perfectly matches the ground truth, and 0 otherwise.

$$r_{\text{EM}} = \begin{cases} 1 & \text{if } y = \hat{y}, \\ 0 & \text{otherwise.} \end{cases} \tag{3}$$

- **F1 Reward** ($r_{\text{F1}}$). Since exact matches occur infrequently in practice, relying solely on EM rewards leads to sparse reward signals. Furthermore, the EM metric fails to distinguish between partially correct answers, as all non-exact matches receive zero reward. Therefore, we introduce F1 reward, which provides a finer-grained evaluation by measuring the degree of partial match.

$$r_{\text{F1}} = \frac{2 \times I_N}{P_N + R_N}, \tag{4}$$

where $P_N$ denotes the number of tokens in the predicted answer, $R_N$ denotes the number of tokens in the gold answer, and $I_N$ is the number of intersecting tokens between the two answers.

The final reward function combines these reward signals through weighted summation:

$$r = r_{\text{EM}} + \alpha \cdot r_{\text{F1}}, \tag{5}$$

where $\alpha \in (0, 1]$ is a hyperparameter that controls the relative contribution of the F1 reward.

### 2.2.4 TRAINING TEMPLATE

Figure 5 displays the prompt employed to train the compressor model for generating a summary of the retrieved documents, conditioned on the given question. This prompt is notably concise. For end-task answer generation, the prompt provided to the LLM $M$ is illustrated in Figure 6, which incorporates few-shot in-context examples, the (generated summary of) retrieved documents, and the question.

### 2.2.5 EFFICIENCY ANALYSIS

**Training Efficiency.** Since our method employs reinforcement learning for training, it incurs greater time and computational costs compared to approaches that do not utilize reinforcement learning (Xu et al., 2024; Cao et al., 2024). However, our training process only optimizes a lightweight compressor model with relatively few parameters, while the larger generator LLM responsible for producing task answers remains fixed and is not updated during training. This design ensures high training efficiency—for instance, training one epoch takes approximately 2 hours using eight H20 GPUs, and convergence is typically achieved within just two epochs. In contrast, other reinforcement learning-based methods, such as ReSearch (Chen et al., 2025a) and R1-Searcher (Song et al., 2025), require direct fine-tuning of the large generator LLM, leading to considerably higher training time and resource consumption. Furthermore, it is important to emphasize that our method exhibits strong generalization capability. As shown in Section 3.3, a model trained only once demonstrates broad applicability, thereby reducing the need for frequent retraining and further lowering the overall training cost.

**Inference Efficiency.** Our method significantly enhances inference efficiency. In contrast to RAG approaches that do not employ a compressor—and thus require the generator LLM to directly encode lengthy documents, often spanning thousands of tokens—our approach introduces a lightweight compressor that processes long documents and summarizes them into compact representations of only a few dozen tokens before feeding them to the generator LLM. Since the compressor is an order of magnitude smaller in parameter size than the generator LLM, it substantially reduces the encoding time that would otherwise be incurred by the generator, leading to notable gains in inference efficiency. It is also important to note that the use of reinforcement learning does not adversely affect inference efficiency, as it is only involved during the training phase.

## 3 EXPERIMENTS

### 3.1 EXPERIMENTAL SETTINGS

**Datasets and Evaluation Metrics.** We evaluate our method on four benchmark datasets: two single-hop question-answering datasets, Natural Questions (NQ) (Kwiatkowski et al., 2019) and TriviaQA (Joshi et al., 2017), as well as two multi-hop question-answering datasets, HotpotQA (Yang et al., 2018) and 2WikiMultihopQA (Ho et al., 2020). Results are reported on the test sets of Natural Questions and TriviaQA, as well as the development sets of HotpotQA and 2WikiMultihopQA. Following RECOMP (Xu et al., 2024), the performance is measured using Exact Match and token-level F1 scores, while efficiency is assessed by the number of tokens provided in the context.

**Compression Model** ($\pi_\theta$). We trained our compression model using Qwen2.5-1.5B-Instruct to generate summaries of the retrieved documents. To evaluate the effect of using different models as compressors, we also trained compressors using Llama3.2-1B-Instruct and Llama3.2-3B-Instruct (Section 3.3).

**Large Language Model** ($M$). We use Qwen2.5-14B-Instruct as the primary LLM model $M$ to generate predicted answers which are used to guide the training of the compressor. To evaluate the generalization ability of our method, we also transfer to another LLM model, LLama3.1-8B-Instruct (Table 2).

Table 1: Open-domain QA results using Qwen2.5-14B-Instruct as the downstream LLM ($M$). The reported token counts represent the length of in-context documents, excluding few-shot examples. RECOMP, NoiseFilter-IB and our method CORE are all trained using Qwen2.5-1.5B-Instruct.

| | NQ | | | TriviaQA | | | HotpotQA | | | 2WikiMultihopQA | | |
|---|---|---|---|---|---|---|---|---|---|---|---|---|
| | EM | F1 | # tok | EM | F1 | # tok | EM | F1 | # tok | EM | F1 | # tok |
| No Retrieval | 21.36 | 30.97 | 0 | 53.23 | 59.98 | 0 | 21.05 | 29.48 | 0 | 26.11 | 29.51 | 0 |
| *RAG without compression* | | | | | | | | | | | | |
| Top1 Document | 34.46 | 44.41 | 142 | 60.82 | 68.70 | 143 | 29.20 | 38.93 | 147 | 26.79 | 31.87 | 153 |
| Top3 Documents | 37.78 | 48.45 | 427 | 62.60 | 71.02 | 430 | 31.64 | 41.87 | 442 | 27.89 | 33.58 | 460 |
| Top5 Documents | 38.03 | 49.16 | 712 | 64.10 | 72.48 | 715 | 32.99 | 43.69 | 737 | 29.64 | 35.21 | 766 |
| Top10 Documents | 38.67 | 50.03 | 1425 | 64.40 | 72.92 | 1428 | 33.95 | 44.88 | 1471 | 31.04 | 36.75 | 1531 |
| *Compression of top 5 documents* | | | | | | | | | | | | |
| BM25 | 25.23 | 36.47 | 37 | 55.36 | 63.90 | 39 | 24.18 | 35.73 | 71 | 25.42 | 30.29 | 68 |
| Qwen2.5-1.5B | 31.94 | 43.03 | 36 | 57.99 | 66.70 | 30 | 27.36 | 37.47 | 33 | 25.93 | 31.18 | 32 |
| DeepSeek-V3 (671B) | 37.73 | 50.39 | 54 | 64.13 | 73.20 | 50 | 33.59 | 44.83 | 48 | 27.99 | 32.67 | 92 |
| RECOMP-Abs (1.5B) | 34.18 | 46.26 | 58 | 60.31 | 68.50 | 53 | 28.96 | 39.95 | 56 | 30.25 | 36.73 | 52 |
| RECOMP-Ext (1.5B) | 33.84 | 46.05 | 56 | 60.18 | 68.39 | 48 | 29.93 | 41.09 | 45 | 30.78 | 37.07 | 51 |
| NoiseFilter-IB (1.5B) | 35.15 | 45.94 | 48 | 59.51 | 68.15 | 35 | 27.97 | 38.62 | 38 | 27.85 | 34.69 | 40 |
| LongLLMLingua (1.5B) | 33.65 | 43.15 | 152 | 58.96 | 66.82 | 148 | 28.03 | 38.49 | 149 | 29.37 | 33.62 | 153 |
| QGC (1.5B) | 36.23 | 45.88 | 49 | 61.02 | 68.45 | 47 | 29.16 | 40.05 | 45 | 31.14 | 36.83 | 51 |
| **CORE (1.5B)** | **41.02** | **50.40** | **46** | **65.63** | **72.55** | **32** | **33.67** | **45.06** | **36** | **36.72** | **42.05** | **49** |
| *Compression of top 10 documents (with the compressor trained on top 5 docs)* | | | | | | | | | | | | |
| BM25 | 25.91 | 36.88 | 38 | 55.28 | 63.16 | 37 | 23.49 | 35.01 | 68 | 25.61 | 30.54 | 65 |
| Qwen2.5-1.5B | 32.94 | 44.84 | 40 | 58.45 | 67.31 | 33 | 28.17 | 38.48 | 36 | 26.22 | 31.57 | 34 |
| DeepSeek-V3 (671B) | 37.79 | 51.07 | 56 | 65.29 | 74.45 | 53 | 34.62 | 45.69 | 50 | 29.00 | 34.64 | 40 |
| RECOMP-Abs (1.5B) | 34.40 | 46.93 | 59 | 61.42 | 69.88 | 52 | 31.54 | 42.92 | 52 | 31.98 | 38.16 | 49 |
| RECOMP-Ext (1.5B) | 33.96 | 46.34 | 60 | 61.03 | 69.51 | 50 | 31.92 | 43.18 | 55 | 32.52 | 38.87 | 44 |
| NoiseFilter-IB (1.5B) | 35.36 | 46.24 | 50 | 59.92 | 68.32 | 38 | 28.21 | 38.83 | 38 | 28.63 | 35.16 | 42 |
| LongLLMLingua (1.5B) | 33.78 | 43.37 | 154 | 59.17 | 66.97 | 150 | 28.33 | 38.95 | 148 | 29.62 | 34.11 | 151 |
| QGC (1.5B) | 36.03 | 45.62 | 50 | 61.23 | 68.74 | 49 | 29.12 | 39.63 | 46 | 31.71 | 37.52 | 50 |
| **CORE (1.5B)** | **41.88** | **51.26** | **52** | **66.76** | **73.64** | **33** | **34.68** | **46.35** | **37** | **37.99** | **43.28** | **48** |

**Retrieval Corpus and Retrievers.** Following previous studies (Xu et al., 2024), we use the Wikipedia corpus from December 20, 2018, as the retrieval source for all four datasets. The articles are segmented into non-overlapping 100-word documents. To ensure that our method is not dependent on a specific retriever, we experiment with several mainstream retrievers. Specifically, we use DPR (Karpukhin et al., 2020) for NQ, a hybrid of DPR and BM25 (Robertson et al., 1995) for TriviaQA, and the Contriever model (Izacard et al., 2021) trained on the MS MARCO dataset (Nguyen et al., 2016) for HotpotQA and 2WikiMultihopQA.

**Baselines.** To evaluate the effectiveness of our method, we compared it against various baselines. First, we evaluated the uncompressed approach—retaining the original in-context RALM setup—by prepending the top 1, 3, 5, and 10 retrieved documents to the prompt. We also tested alternative compression methods, including the traditional BM25 algorithm (which ranks sentences by their similarity to the input), off-the-shelf Qwen2.5-1.5B-Instruct model (with comparable parameter size to our approach), and DeepSeek-V3 model (671B parameters, far exceeding our compressor's capacity). Furthermore, we included state-of-the-art context compression methods for RAG, RECOMP (Xu et al., 2024), NoiseFilter-IB (Zhu et al., 2024), LongLLMLingua (Jiang et al., 2024) and QGC (Cao et al., 2024). For RECOMP, we evaluated both its abstractive and extractive variants. To ensure a fair comparison, all trainable approaches were all trained using the same model.

**Implementation Details.** For the distillation warm-up phase, we perform full-parameter supervised fine-tuning on the off-the-shell language model for two epochs using LLaMA-Factory [1]. This warmed-up model then serves as the initializer for the subsequent reinforcement learning phase. We adopt the Verl framework [2] for RL training. The initialized compression model is trained for two epochs on each dataset. Training is conducted on eight NVIDIA H20 GPUs using full parameter GRPO optimization, with a learning rate of 1e-5, a batch size of 256, five rollouts per sample, and a KL loss coefficient of 0.001. The downstream LLM ($M$) used for reward generation is served using the vLLM inference engine during RL training.

---

[1]https://github.com/hiyouga/LLaMA-Factory

[2]https://github.com/volcengine/verl

## 3.2 OVERALL PERFORMANCE

The detailed comparison results are presented in Table 1. We evaluate the following approaches: traditional RAG without compression using the top 1, 3, 5, and 10 documents prepended to the original input; BM25-based compression; off-the-shelf LLMs (Qwen2.5-1.5B-Instruct and DeepSeek-V3); and state-of-the-art trainable RAG compressors (RECOMP-Abstractive, RECOMP-Extractive, and NoiseFilter-IB). For fair comparison, all trainable methods were trained using the same backbone model, Qwen2.5-1.5B-Instruct. These compressors were trained on five-document inputs, and we report their performance both on in-domain five-document compression and out-of-domain generalization to ten-document compression.

**RAG versus No RAG.** As shown in Table 1, prepending the original input with retrieved documents yields a substantial improvement over the no-retrieval baseline, indicating that these documents provide valuable information for generating the answer. Performance improves as the number of retrieved documents increases from 1 to 10, albeit with diminishing returns—a trend attributable to the decreasing relevance of lower-ranked documents. However, this performance gain comes at the cost of a significant increase in the number of additional tokens the LLM must encode, rising from 0 to over 1,400 tokens.

**Lossless Compression of CORE.** Our compressor was trained using a context of five documents. As presented in Table 1, compared to prepending the full content of all five documents to the original input (i.e., the uncompressed baseline), our method achieves a high compression rate of approximately 6% *with no loss in performance*. Remarkably, on all four datasets, the approach not only maintains performance but also enhances EM by 1 to 7 points. For example, on NQ, the model achieves an EM of 41.02 with compressed input, surpassing the uncompressed score of 38.03.

**Comparison with Compression Baselines.** When compressing the top five documents, all baseline compression methods result in performance degradation to varying degrees compared to the uncompressed baseline. Specifically, BM25 leads to a substantial performance drop. Using the off-the-shelf Qwen2.5-1.5B-Instruct model for compression yields better results than BM25 but still underperforms relative to the uncompressed baseline. Surprisingly, even the large-scale DeepSeek-V3 model (with 671B parameters) achieves only near-lossless compression on TriviaQA and HotpotQA, while performance on NQ and 2Wiki remains below the uncompressed baseline. As for the trained compressors (RECOMP, NoiseFilter-IB, LongLLMLingua, and QGC), all of them exhibit performance degradation compared to no compression. The decline generally ranges from 2 to 6 EM points across nearly all datasets. In contrast, our method, CORE, delivers the best performance. It not only surpasses compression methods of comparable size by 4-5 EM points but also maintains an advantage over the hundreds-of-times-larger DeepSeek model. This clearly demonstrates the benefit and importance of end-to-end optimization.

**Generalization to 10-Doc Compression.** When generalizing the trained compressors to handle the top-10 documents without retraining, the aforementioned conclusions remain valid. **CORE** continues to achieve lossless compression and performs best among all compression methods. On NQ, it achieves a token compression ratio of 3.6% while improving the EM by 3.2 points compared to using all ten documents. Similar trends are observed on TriviaQA, with a compression ratio of 2.3% and a gain of 2.4 EM points relative to the full-document baseline. On HotpotQA, CORE achieves a compression ratio of 2.5% and an improvement of 0.7 EM points. For 2WikiMultihopQA, it obtains a compression ratio of 3.1% along with a notable increase of 6.9 EM points.

## 3.3 ROBUSTNESS AND GENERALIZATION ABILITY ANALYSIS

**Robustness of CORE Across Compressor Architectures.** To evaluate whether the effectiveness of CORE depends on the choice of compressor model, we compared the performance of compressors trained using different model architectures—while keeping the downstream LLM (Qwen2.5-14B-Instruct) fixed. The compressors tested include LLaMA3.2-1B-Instruct, Qwen2.5-1.5B-Instruct, and LLaMA3.2-3B-Instruct, which vary in architecture and parameter count. As shown in Figure 3, the results indicate that: (1) These trained compressors consistently achieve lossless compression and outperform the uncompressed baseline (represented by the red reference line in the figure, which corresponds to prepending the full document content), confirming that our training framework is robust and not tied to a specific compressor architecture. (2) Compression performance improves as the size of the compressor model increases, consistent with scaling laws. More detailed results can be found in Tables 7 and 8.

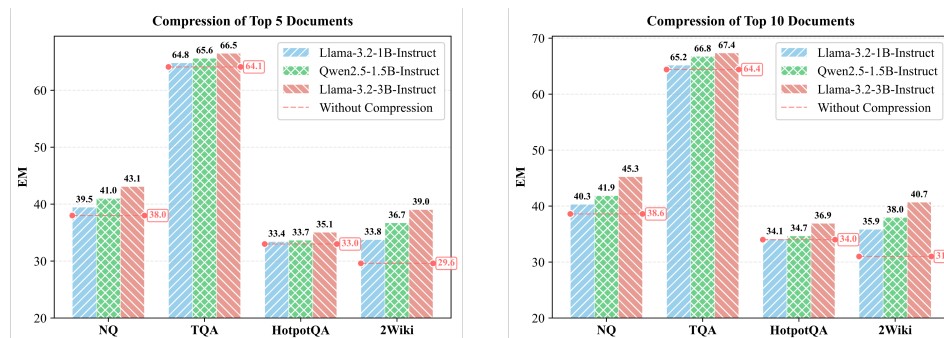

Figure 3: The impact of different models used to train the compressor.

Table 2: Zero-shot transfer of the the trained compressor to Llama-3.1-8B-Instruct.

| | NQ | | | TriviaQA | | | HotpotQA | | | 2WikiMultihopQA | | |
|---|---|---|---|---|---|---|---|---|---|---|---|---|
| | EM | F1 | # tok | EM | F1 | # tok | EM | F1 | # tok | EM | F1 | # tok |
| No Retrieval | 24.04 | 34.91 | 0 | 55.64 | 62.57 | 0 | 19.93 | 27.75 | 0 | 27.64 | 31.18 | 0 |
| *RAG without compression* | | | | | | | | | | | | |
| Top1 Document | 33.80 | 44.06 | 142 | 59.17 | 67.50 | 143 | 27.95 | 37.49 | 147 | 28.41 | 33.43 | 153 |
| Top3 Documents | 36.87 | 47.81 | 427 | 61.13 | 70.06 | 430 | 30.17 | 40.71 | 442 | 28.67 | 34.23 | 460 |
| Top5 Documents | 37.65 | 48.87 | 712 | 62.26 | 71.04 | 715 | 31.44 | 42.16 | 737 | 29.43 | 35.18 | 766 |
| Top10 Documents | 38.12 | 49.93 | 1425 | 63.95 | 72.71 | 1428 | 32.19 | 42.62 | 1471 | 30.45 | 36.04 | 1531 |
| *Compression of top 5 documents* | | | | | | | | | | | | |
| Qwen2.5-1.5B | 32.60 | 44.21 | 36 | 56.76 | 65.77 | 30 | 26.86 | 36.90 | 33 | 25.45 | 30.88 | 32 |
| DeepSeek-V3 (671B) | 37.56 | 50.11 | 54 | 62.52 | 72.34 | 50 | 33.05 | 44.25 | 48 | 28.64 | 33.87 | 92 |
| RECOMP-Abs (1.5B) | 33.41 | 45.50 | 58 | 58.50 | 67.37 | 53 | 28.85 | 39.76 | 56 | 31.63 | 37.81 | 52 |
| RECOMP-Ext (1.5B) | 33.12 | 45.06 | 60 | 57.98 | 66.84 | 55 | 29.03 | 40.04 | 52 | 31.85 | 38.02 | 55 |
| **CORE (1.5B)** | **40.72** | **50.00** | **46** | **64.08** | **71.13** | **32** | **32.17** | **43.71** | **36** | **35.99** | **41.42** | **49** |
| *Compression of top 10 documents* | | | | | | | | | | | | |
| Qwen2.5-1.5B | 32.88 | 44.66 | 40 | 57.44 | 66.56 | 33 | 27.31 | 37.31 | 36 | 25.80 | 31.30 | 34 |
| DeepSeek-V3 (671B) | 37.49 | 51.28 | 56 | 63.79 | 73.80 | 53 | 34.24 | 45.35 | 50 | 31.45 | 37.09 | 40 |
| RECOMP-Abs (1.5B) | 34.18 | 46.80 | 59 | 59.69 | 68.89 | 52 | 30.17 | 41.42 | 55 | 33.61 | 39.78 | 44 |
| RECOMP-Ext (1.5B) | 34.06 | 46.55 | 60 | 59.33 | 68.71 | 50 | 30.52 | 41.98 | 55 | 33.52 | 39.42 | 44 |
| **CORE (1.5B)** | **41.77** | **51.27** | **52** | **65.25** | **72.45** | **33** | **33.25** | **45.09** | **37** | **37.59** | **42.87** | **48** |

**Transferability of CORE Across Downstream LLMs.** We evaluate the transferability of our trained compressor and other baseline compressors to a new downstream LLM, the LLaMA-3.1-8B model. The results are presented in Table 2. Note that all trainable compressors, including our own, were trained using feedback generated by Qwen2.5-14B-Instruct. The findings reveal that existing trained compressors (e.g., RECOMP) exhibit limited generalization capability, as indicated by a larger performance gap relative to the baseline of prepending full documents. In contrast, CORE demonstrates stronger generalization, achieving lossless compression on the new downstream LLM. Specifically, it not only maintains performance but also surpasses the full-document baseline (i.e., no compression) across all four datasets while retaining a high compression rate. These results suggest that the summaries produced by our method are inherently high-quality and preserve key information necessary for accurate answering, thereby enabling effective transfer to other LLMs.

## 3.4 ABLATION STUDY

Table 3 presents an ablation study on the two stages of our method: distillation and GRPO. Here, "w/o distillation" denotes training the compressor with GRPO directly, bypassing the warm-start phase, while "w/o GRPO" indicates using only the distillation step without subsequent reinforcement learning. The results demonstrate that removing either component leads to performance degradation, confirming the necessity of both. In addition, the decline is more substantial when GRPO is omitted, highlighting the crucial role of reinforcement learning in the absence of explicit supervision. Distillation injects external knowledge into the model, providing a favorable starting point for RL training and thereby enabling RL to more effectively unlock the compressor's full potential.

Table 3: Ablation study.

| Dataset | Metric | w/o distillation | w/o GRPO | CORE |
|---------|--------|------------------|----------|------|
| NQ | EM | 36.37 | 34.18 | **41.02** |
|  | F1 | 46.91 | 46.26 | **50.40** |
| TQA | EM | 65.23 | 60.31 | **65.63** |
|  | F1 | 72.41 | 68.50 | **72.55** |
| HotpotQA | EM | 32.01 | 28.96 | **33.67** |
|  | F1 | 42.73 | 39.95 | **45.06** |
| 2Wiki | EM | 31.40 | 30.25 | **36.72** |
|  | F1 | 36.89 | 36.73 | **42.05** |

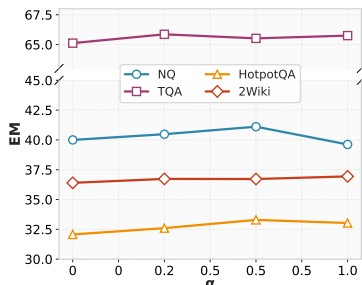

Figure 4: The impact of $\alpha$.

### 3.5 THE IMPACT OF $\alpha$.

Figure 4 illustrates the performance of our method under different values of $\alpha$, which controls the weighting coefficient of the F1 reward term. Setting $\alpha = 0$ corresponds to using only the EM reward. Results indicate that performance improves when $\alpha > 0$ across all datasets, demonstrating the effectiveness of the F1 reward in mitigating the sparsity issue associated with the EM reward. However, the optimal value of $\alpha$ is dataset-dependent; values between 0.2 and 0.5 generally yield strong performance.

### 3.6 CASE STUDY

To conduct an in-depth analysis of the advantages of our compressor, we performed case studies on one single-hop QA dataset (NQ) and one multi-hop QA dataset (2Wiki), with the results presented in Table 4 and Table 5, respectively. For each case, we compared the summaries generated by off-the-shelf Qwen2.5-1.5B-Instruct, RECOMP, and our method CORE based on the same set of documents, as well as the predicted answers generated by the LLM after prepending these summaries. As shown in the tables, although the summaries produced by off-the-shelf Qwen2.5-1.5B are concise, they largely fail to capture key information relevant to answering the question. In contrast, RECOMP demonstrates better summarization capability but is prone to being overwhelmed by lengthy documents, resulting in misjudgments and even generating misleading information—such as the statement in Table 4: *"The U.S. stopped drafting for the Vietnam War after the Selective Service System was officially abolished in December 1972"*—which leads the downstream LLM to produce the incorrect answer "1972". Our method, CORE, accurately extracts answer-critical information from lengthy documents, exemplified by the summary: *"The U.S. stopped drafting for the Vietnam War in 1973 after announcing the decision by Secretary of Defense Melvin Laird earlier that year"*, thereby enabling the LLM to generate the correct answer "1973". This indicates that our compressor, trained with an end-task target-oriented reinforcement learning optimization strategy, can produce document summaries that are most helpful for answering the given question while effectively filtering out irrelevant information.

## 4 RELATED WORK

**Context Compression in RAG.** RAG enhances the performance of LLMs on knowledge-intensive tasks by retrieving the most relevant documents from extensive knowledge bases and prepending them as contextual information to the original input (Ram et al., 2023; Fan et al., 2024; Lin et al., 2023; Shi et al., 2023). However, this approach requires the LLM to process significantly longer token sequences, resulting in increased computational costs. To mitigate this issue, researchers have begun to explore methods for compressing retrieved documents in RAG systems (Xu et al., 2024; Cao et al., 2024; Jin et al., 2024b; Zhu et al., 2024; Kim & Thorne, 2025; Rau et al., 2024; Wu et al., 2025; Louis et al., 2025; Jin et al., 2024a; Li et al., 2024a;b; Zhang et al., 2024a). For instance, Xu et al. (2024) propose compressing retrieved documents into textual summaries before in-context augmentation, training the compressor through data selection and distillation. Similarly, Cao et al. (2024) introduce a Query-Guided Compressor (QGC) that uses queries to guide the compression process, effectively preserving essential information. Jin et al. (2024b) refine retrieved documents into Key Supporting Evidence (KSE) through a combination of knowledge synthesis, supervised fine-tuning (SFT), and preference alignment. Meanwhile, Zhu et al. (2024) present an information-theoretic approach called NoiseFilter-IB, which filters noise by maximizing the mutual information

between the compressed content and the ground-truth output. Additionally, Kim & Thorne (2025) train a compressor to extract critical information using reward functions based on predefined heuristic rules. However, most of these methods are heuristic in nature, and due to the lack of ideal compression labels, the compressed content they produce cannot be guaranteed to benefit downstream LLMs. In contrast, our method, CORE, adopts an end-to-end optimization approach to address these limitations.

**Reinforcement Learning.** Reinforcement learning (RL) has recently achieved notable success, enabling LLMs to develop reasoning capabilities without explicit step-by-step supervision (Liu et al., 2024; Shao et al., 2024; Guo et al., 2025). Building on these advances, several studies have applied RL to improve RAG (Ke et al., 2024). For example, Kulkarni et al. (2024) use RL to autonomously decide whether to retrieve documents, while Zhang et al. (2024b) employ RL to optimize the ranking of retrieved documents. Similarly, Mao et al. (2024) propose a framework for training query rewriting models for RAG without relying on human annotations. MMOA-RAG (Chen et al., 2025b) enhances RAG through multi-agent reinforcement learning, incorporating a query rewriter, retriever, and generator. Meanwhile, RL has been applied to address other challenges in RAG, such as enhancing the quality of retrieved content. For instance, Oreo (Li & Ramakrishnan, 2025) trains a reconstructor with PPO and ROUGE-based rewards to rewrite passages for improved performance. In contrast, our work CORE tackles the distinct problem of computational efficiency. Our approach leverages the GRPO algorithm, direct task-performance rewards, and a lightweight compressor to enhance efficiency without compromising accuracy. Moreover, a line of research has utilized RL to integrate search with reasoning in a step-by-step manner (Singh et al., 2025). For instance, Chen et al. (2025a) introduce a framework called ReSearch, which trains LLMs to reason with search using RL, without requiring supervised data for reasoning steps. Related approaches include R1-Searcher (Song et al., 2025), WebThinker (Li et al., 2025), and DeepResearcher (Zheng et al., 2025). Although these methods are end-to-end, they differ fundamentally from our problem setting. These approaches typically involve directly training the LLM generator—which tends to be a large-scale model with a high parameter count. However, such training becomes infeasible when the model is a black box (e.g., GPT-4), as internal weights or gradients are inaccessible. Furthermore, these methods introduce extensive internal thinking processes that substantially increase context length and inference time. In contrast, our approach treats the generator LLM as a fixed black-box model and trains only a lightweight plug-in compressor to produce document summaries. This design significantly improves both training and inference efficiency. A parallel line of work applies RL to prompt compression. For example, PCRL (Jung & Kim, 2024) and TACO-RL (Shandilya et al., 2025) learn to compress prompts via token-level keep-or-drop decisions, using the similarity between model outputs with compressed and original prompts as the reward. Our work, CORE, introduces key distinctions in both objective and methodology. First, we target the more complex problem of compressing multiple retrieved documents in RAG, rather than single prompts. Second, we employ a generative compressor that can rephrase and synthesize content, instead of making token-level binary actions. Most importantly, we optimize compression using a direct task-performance reward with GRPO, which enables us to achieve true lossless compression at significantly higher ratios—a stark contrast to the performance degradation observed in prior prompt compression methods.

## 5 CONCLUSION

This paper analyzes the limitations of current context compression methods for RAG. A primary challenge is the lack of optimal reference summaries for supervised learning, which often results in performance degradation in downstream tasks. To overcome this, we frame context compression as a reinforcement learning problem, utilizing downstream task performance as a reward signal to train the compression policy, thereby enabling end-to-end optimization. Extensive experiments demonstrate that our proposed method, CORE, achieves effectively lossless compression by maintaining a high compression ratio while preserving original task performance. Surprisingly, CORE not only preserves but actually enhances performance on all downstream tasks. Further in-depth analysis provides additional insights into the efficacy of our approach.

ETHICS STATEMENT

The authors affirm that this work adheres to the ICLR Code of Ethics. It involves no human subjects, sensitive or private data, or applications posing potential ethical risks. All resources utilized are publicly available and appropriately licensed. The research was conducted in accordance with ethical and legal standards.

REPRODUCIBILITY STATEMENT

This paper includes detailed descriptions of the experimental setups, implementation details, hyperparameter selections, and evaluation procedures to facilitate full verification of the reported results. To further support reproducibility, the complete source code and experimental scripts are available at the following anonymous repository: `https://anonymous.4open.science/r/CORE-28B4`.

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

> Compressed documents:

Figure 5: Prompt template used to generate the summary.

> [Instruction] Answer the question.
> IMPORTANT: Respond ONLY with the exact answer in the same format as the examples. Do NOT add any extra text, explanations, or punctuation. Do NOT include "Answer:" or any similar prefix in your response.
>
> [Examples]
> Question: Which major Russian city borders the body of water in which Saaremaa is located?
> Answer: Saint Petersburg
> Question: Who was thee first president of the association that wrote the code of ethics for psychology?
> Answer: G. Stanley Hall
> Question: Where did the Baldevins bryllup director die?
> Answer: Copenhagen
>
> [Current Question]
> {***Summary of the retrieved documents***}
> Question: {*Question*}
> Answer:

Figure 6: Prompt template for LLM QA.

## A    USE OF LLMS

Large language models (LLMs) were employed solely as writing assistants to enhance the language, improve clarity, and check grammatical correctness. They were not used to generate research ideas, design or implement methodologies, conduct data analysis, or produce any of the results presented in this work. The authors assume full responsibility for the entire content of the paper.

## B    PROMPT TEMPLATES

Figure 5 displays the prompt employed to train the compressor model for generating a summary of the retrieved documents, conditioned on the given query. This prompt is notably concise. For end-task answer generation, the prompt provided to the LLM $M$ is illustrated in Figure 6, which incorporates few-shot in-context examples, the (generated summary of) retrieved documents, and the question.

## C    CASE STUDY

To conduct an in-depth analysis of the advantages of our compressor, we performed case studies on one single-hop QA dataset (NQ) and one multi-hop QA dataset (2Wiki), with the results presented in Table 4 and Table 5, respectively. For each case, we compared the summaries generated by off-the-shelf Qwen2.5-1.5B-Instruct, RECOMP, and our method CORE based on the same set of documents, as well as the predicted answers generated by the LLM after prepending these summaries. As shown in the tables, although the summaries produced by off-the-shelf Qwen2.5-1.5B

Table 4: Case study on NQ dataset.

**Question: when did the us stop drafting for the vietnam war? Gold answer: [1973]**

**Top-5 documents**:

last men conscripted, who were born in 1952 and who reported for duty in June 1973. On February 2, 1972, a drawing was held to determine draft priority numbers for men born in 1953, but in early 1973 it was announced by Secretary of Defense Melvin Laird that no further draft orders would be issued. In March 1973, 1974, and 1975, the Selective Service assigned draft priority numbers for all men born in 1954, 1955, and 1956, in case the draft was extended, but it never was. Command Sergeant Major Jeff Mellinger, believed to be the last drafted enlisted ranked.

The Gates Commission issued its report in February 1970, describing how adequate military strength could be maintained without having conscription. The existing draft law was expiring at the end of June 1971, but the Department of Defense and Nixon administration decided the draft needed to continue for at least some time. In February 1971, the administration requested of Congress a two-year extension of the draft, to June 1973. Senatorial opponents of the war wanted to reduce this to a one-year extension, or eliminate the draft altogether, or tie the draft renewal to a timetable for troop withdrawal from Vietnam; Senator.

draft operated from 1940 until 1946 when further inductions were suspended, and its legislative authorization expired without further extension by Congress in 1947. During this time, more than 10 million men had been inducted into military service. However, the Selective Service System remained intact. Scattered opposition was encountered especially in the northern cities where some African-Americans protested the system. The tiny Nation of Islam was at the forefront, with many Black Muslims jailed for refusing the draft, and their leader Elijah Muhammed was sentenced to federal prison for 5 years for inciting draft resistance. Organized draft resistance also developed in.

was the first time any form of student deferment was used. During the Korean War a student carrying at least twelve semester hours was spared until the end of his current semester. The United States breathed easier with the Korean War Armistice on July 27, 1953; however, technology brought new promises and threats. U.S. air and nuclear power fueled the Eisenhower doctrine of "massive retaliation". This strategy demanded more machines and fewer foot soldiers, so the draft slipped to the back burner. However, the head of the SSS, Maj. Gen. Hershey, urged caution fearing the conflict looming in Vietnam. In.

policy of using the draft as force to compel "voluntary" enlistment was unique in U.S. history. Previous drafts had not aimed at encouraging individuals to sign up in order to gain preferential placement or less dangerous postings. However, the incremental buildup of Vietnam without a clear threat to the country bolstered this. Some estimates suggest conscription encompassed almost one-third of all eligible men during the period of 1965–69. This group represented those without exemption or resources to avoid military service. During the active combat phase, the possibility of avoiding combat by selecting their service and military specialty led as many.

| Compressor | Generated summary | Predicted answer |
|---|---|---|
| **Off-the-shell Qwen2.5-1.5B-Instruct** | The US stopped drafting for the Vietnam War in late 1972 after the draft expired at the end of June 1971. | late 1972 |
| **RECOMP** | The U.S. stopped drafting for the Vietnam War after the Selective Service System was officially abolished in December 1972 due to growing public opposition and the expiration of the draft's legal authority in 1971. While the draft was formally ended, some individuals were still consripted into the military | December 1972 |
| **Ours** | The U.S. stopped drafting for the Vietnam War in **1973** after announcing the decision by Secretary of Defense Melvin Laird earlier that year. Although the Selective Service System was later updated to include draft priority numbers, this decision marked the end of the draft's use for national service. | 1973 |

Table 5: Case study on 2Wiki dataset.

**Question: Who is Charles Bretagne Marie De La Trémoille's paternal grandfather?**
**Gold answer: [Charles Armand René de La Trémoille]**

**Top-5 documents**:

as at Versailles: he was brigadier of cavalry (January 1709), first gentleman of the King's chamber (June 1709), governor of Thouars (July 1709), and Maréchal de camp (February 1719). His sister Marie Armande Victoire de La Trémoille married Emmanuel Théodose de La Tour d'Auvergne. On 13 April 1706 he married Marie-Madeleine Motier de La Fayette (1691–1717), the daughter of Rene-Armand, marquis de La Fayette and Marie-Madeleine de Marillac, and granddaughter of the author Marie-Madeleine Pioche de la Vergne, comtesse de la Fayette. They had one child, Charles Armand René de La Trémoille, born in 1708. Charles Louis Bretagne de La

Charles Bretagne Marie de La Trémoille Charles Bretagne Marie de La Trémoille (24 March 1764 – 10 November 1839), 8th duc de Thouars, was a French soldier and the son of Jean Bretagne Charles de La Trémoille and his wife, Marie-Maximilienne, princess of Salm-Kyrburg. La Trémoille married Louise-Emmanuelle de Châtillon in 1781. She was a grand daughter of Louis César de La Baume Le Blanc, the famous writer. The couple had one daughter: At the outbreak of the French Revolution, he sided with King Louis XVI, as did the rest of his family. In 1789, La Trémoille and his parents

Jean Bretagne Charles de La Trémoille Jean-Bretagne-Charles de La Trémoille (5 February 1737 – 19 May 1792), 7th duc de Thouars, was a French soldier and the son of Charles Armand René de La Trémoille and his wife, Marie Hortense de La Tour d'Auvergne. La Trémoille began his military career in the Seven Years' War as the head of the regiment of Aquitaine-Cavalry. He was seriously wounded and afterwards promoted to the rank of sergeant, and later of brigadier, of the armies of the king. During the French Revolution, La Trémoille remained faithful to King Louis XVI. He emigrated in

Charles Louis Bretagne de La Trémoille Charles Louis Bretagne de La Trémoille (15 March 1683 – 9 October 1719), 6th duke of Thouars, was the son of Charles Belgique Hollande de La Trémoille and Madeleine de Créquy, daughter and heiress of Charles III de Créquy. He became duke of Thouars upon his father's death in 1709. He was also count of Laval and Montfort and inherited the ancient French claim to the Kingdom of Naples. The Neapolitan claim gave the family the rank of "princes étrangers" at the French court. The Duke made a career in the military as well

de Saint-Didier. They had two daughters: Following her death, La Trémoille married for a third time in 1830 to Valentine Eugénie Joséphine Walsh de Serrant. They had two children: La Trémoille died shortly after his son's birth, in 1839. Charles Bretagne Marie de La Trémoille Charles Bretagne Marie de La Trémoille (24 March 1764 – 10 November 1839), 8th duc de Thouars, was a French soldier and the son of Jean Bretagne Charles de La Trémoille and his wife, Marie-Maximilienne, princess of Salm-Kyrburg. La Trémoille married Louise-Emmanuelle de Châtillon in 1781. She was a grand daughter of Louis César de

| Compressor | Generated summary | Predicted answer |
|---|---|---|
| **Off-the-shell Qwen2.5-1.5B-Instruct** | Charles Bretagne Marie de La Trémoille's paternal grandfather was Jean Bretagne Charles de La Trémoille. | Jean Bretagne Charles de La Trémoille |
| **RECOMP** | Charles Bretagne Marie de La Trémoille's paternal grandfather was Charles Louis Bretagne de La Trémoille, who was the son of Charles Belgique Hollande de La Trémoille and Madeleine de Créquy. This makes him the direct paternal grandfather of Charles Bretagne Marie de La Trémoille. | Charles Louis Bretagne de La Trémoille |
| **Ours** | Charles Bretagne Marie de La Trémoille's paternal grandfather is Charles Armand René de La Trémoille, her father's father, the 7th duc de Thouars. | Charles Armand René de La Trémoille |

Table 6: Failure Cases on NQ dataset.

| Question | Generated Summary | Predicted Answer | Gold Answer |
|---|---|---|---|
| who is the president of usa right now | The current president of the USA is **Donald Trump**, who was elected on January 20, 2017, and is the 45th president. He is a businessman and reality TV personality, and his inauguration marked a shift in the country's political landscape. | Joe Biden | Donald Trump |
| who has been ranked no. 1 in the latest football rankings announced by fifa | The latest FIFA football rankings, announced in 2018, have ranked France as the top team, after winning the 2018 World Cup. This change was implemented using the Elo rating system, and the rankings were introduced to better reflect football team strengths. | France | Germany |

are concise, they largely fail to capture key information relevant to answering the question. In contrast, RECOMP demonstrates better summarization capability but is prone to being overwhelmed by lengthy documents, resulting in misjudgments and even generating misleading information—such as the statement in Table 4: "*The U.S. stopped drafting for the Vietnam War after the Selective Service System was officially abolished in December 1972*"—which leads the downstream LLM to produce the incorrect answer "1972". Our method, CORE, accurately extracts answer-critical information from lengthy documents, exemplified by the summary: "*The U.S. stopped drafting for the Vietnam War in 1973 after announcing the decision by Secretary of Defense Melvin Laird earlier that year*", thereby enabling the LLM to generate the correct answer "1973". This indicates that our compressor, trained with an end-task target-oriented reinforcement learning optimization strategy, can produce document summaries that are most helpful for answering the given question while effectively filtering out irrelevant information.

To further understand the limitations of our approach, we present two failure cases from the NQ dataset where the model provided incorrect answers based on our generated summaries. As shown in Table 6, the first case reveals that although the summary contained the key information required for the correct answer, the downstream LLM still produced an error, potentially due to its over-reliance on parametric knowledge. In the second case, the summary itself omitted critical information needed to answer the question, which likely led to the incorrect response.

## D    IMPACT OF DIFFERENT COMPRESSORS ON PERFORMANCE

In our previous experiments, we employed Qwen2.5-1.5B as the initial model to train our compressor. In this section, we utilize two additional models—Llama3.2-1B and Llama3.2-3B—as starting points to train our compressor and the baseline compressor, respectively. The experimental results are presented in Table 7 and Table 8. As shown in the results, our method CORE continues to achieve lossless compression with both models, maintaining a high token compression ratio while exhibiting no performance degradation in terms of Exact Match (EM) and F1 score compared to uncompressed RAG. Furthermore, under both new model configurations, our approach consistently outperforms the baseline methods, indicating that its superiority is not dependent on a specific model architecture and thus demonstrates strong robustness.

We also observe that our method adheres to a form of scaling law: the compressor trained using the 3B model outperforms the one trained with the 1B model. Specifically, the 1B compressor improves performance by 1–4 EM points over the uncompressed baseline, while the 3B compressor yields gains of 3–9 EM points.

Table 7: Open-domain QA results using Qwen2.5-14B-Instruct as the downstream LLM ($M$). The reported token counts represent the length of in-context documents, excluding few-shot examples. RECOMP and our method CORE are both trained using **llama3.2-1B-Instruct**.

| | NQ | | | TriviaQA | | | HotpotQA | | | 2WikiMultihopQA | | |
|---|---|---|---|---|---|---|---|---|---|---|---|---|
| | EM | F1 | # tok | EM | F1 | # tok | EM | F1 | # tok | EM | F1 | # tok |
| No Retrieval | 0.2136 | 0.3097 | 0 | 0.5323 | 0.5998 | 0 | 0.2105 | 0.2948 | 0 | 0.2611 | 0.2951 | 0 |
| *RAG without compression* | | | | | | | | | | | | |
| Top1 Document | 0.3446 | 0.4441 | 142 | 0.6082 | 0.6870 | 143 | 0.2920 | 0.3893 | 147 | 0.2679 | 0.3187 | 153 |
| Top3 Documents | 0.3778 | 0.4845 | 427 | 0.6260 | 0.7102 | 430 | 0.3164 | 0.4187 | 442 | 0.2789 | 0.3358 | 460 |
| Top5 Documents | 0.3803 | 0.4916 | 712 | 0.6410 | 0.7248 | 715 | 0.3299 | 0.4369 | 737 | 0.2964 | 0.3521 | 766 |
| Top10 Documents | 0.3867 | 0.5003 | 1425 | 0.6440 | 0.7292 | 1428 | 0.3395 | 0.4488 | 1471 | 0.3104 | 0.3675 | 1531 |
| *Compression of top 5 docs* | | | | | | | | | | | | |
| llama3.2-1B | 0.3147 | 0.4227 | 64 | 0.5552 | 0.6415 | 60 | 0.2648 | 0.3639 | 58 | 0.2498 | 0.3003 | 61 |
| Deepseek-V3 (671B) | 0.3773 | 0.5039 | 54 | 0.6528 | 0.7433 | 51 | 0.3359 | 0.4483 | 48 | 0.2507 | 0.3031 | 45 |
| RECOMP (1B) | 0.3410 | 0.4655 | 57 | 0.6071 | 0.6880 | 48 | 0.2987 | 0.4121 | 49 | 0.3045 | 0.3653 | 33 |
| **CORE (1B)** | **0.3947** | **0.4923** | **47** | **0.6483** | **0.7287** | **43** | **0.3344** | **0.4454** | **45** | **0.3378** | **0.3969** | **34** |
| *Compression of top 10 docs (with the compressor trained on top 5 docs)* | | | | | | | | | | | | |
| llama3.2-1B | 0.3141 | 0.4228 | 62 | 0.5651 | 0.6512 | 58 | 0.2663 | 0.3661 | 56 | 0.2493 | 0.3006 | 61 |
| Deepseek-V3 (671B) | 0.3779 | 0.5107 | 56 | 0.6529 | 0.7445 | 53 | 0.3462 | 0.4569 | 50 | 0.2900 | 0.3464 | 40 |
| RECOMP (1B) | 0.3421 | 0.4661 | 59 | 0.6095 | 0.6917 | 52 | 0.2982 | 0.4105 | 55 | 0.3072 | 0.3681 | 44 |
| **CORE (1B)** | **0.4033** | **0.5033** | **47** | **0.6521** | **0.7296** | **45** | **0.3412** | **0.4500** | **48** | **0.3586** | **0.4162** | **42** |

Table 8: Open-domain QA results using Qwen2.5-14B-Instruct as the downstream LLM ($M$). The reported token counts represent the length of in-context documents, excluding few-shot examples. RECOMP and our method CORE are both trained using **llama3.2-3B-Instruct**.

| | NQ | | | TriviaQA | | | HotpotQA | | | 2WikiMultihopQA | | |
|---|---|---|---|---|---|---|---|---|---|---|---|---|
| | EM | F1 | # tok | EM | F1 | # tok | EM | F1 | # tok | EM | F1 | # tok |
| No Retrieval | 0.2136 | 0.3097 | 0 | 0.5323 | 0.5998 | 0 | 0.2105 | 0.2948 | 0 | 0.2611 | 0.2951 | 0 |
| *RAG without compression* | | | | | | | | | | | | |
| Top1 Document | 0.3446 | 0.4441 | 142 | 0.6082 | 0.6870 | 143 | 0.2920 | 0.3893 | 147 | 0.2679 | 0.3187 | 153 |
| Top3 Documents | 0.3778 | 0.4845 | 427 | 0.6260 | 0.7102 | 430 | 0.3164 | 0.4187 | 442 | 0.2789 | 0.3358 | 460 |
| Top5 Documents | 0.3803 | 0.4916 | 712 | 0.6410 | 0.7248 | 715 | 0.3299 | 0.4369 | 737 | 0.2964 | 0.3521 | 766 |
| Top10 Documents | 0.3867 | 0.5003 | 1425 | 0.6440 | 0.7292 | 1428 | 0.3395 | 0.4488 | 1471 | 0.3104 | 0.3675 | 1531 |
| *Compression of top 5 docs* | | | | | | | | | | | | |
| llama3.2-3B | 0.3252 | 0.4334 | 60 | 0.5650 | 0.6521 | 59 | 0.2772 | 0.3809 | 58 | 0.2485 | 0.2995 | 60 |
| Deepseek-V3 (671B) | 0.3773 | 0.5039 | 54 | 0.6528 | 0.7433 | 51 | 0.3359 | 0.4483 | 48 | 0.2507 | 0.3031 | 45 |
| RECOMP (3B) | 0.3657 | 0.4912 | 55 | 0.6183 | 0.6920 | 47 | 0.3025 | 0.4238 | 52 | 0.3274 | 0.3806 | 42 |
| **CORE (3B)** | **0.4310** | **0.5234** | **32** | **0.6650** | **0.7306** | **38** | **0.3507** | **0.4736** | **51** | **0.3905** | **0.4474** | **40** |
| *Compression of top 10 docs (with the compressor trained on top 5 docs)* | | | | | | | | | | | | |
| llama3.2-3B | 0.3318 | 0.4359 | 61 | 0.5720 | 0.6588 | 57 | 0.2791 | 0.3854 | 60 | 0.2491 | 0.3011 | 59 |
| Deepseek-V3 (671B) | 0.3779 | 0.5107 | 56 | 0.6529 | 0.7445 | 53 | 0.3462 | 0.4569 | 50 | 0.2900 | 0.3464 | 40 |
| RECOMP (3B) | 0.3682 | 0.4963 | 52 | 0.6205 | 0.6973 | 44 | 0.3077 | 0.4261 | 54 | 0.3312 | 0.3869 | 50 |
| **CORE (3B)** | **0.4526** | **0.5467** | **33** | **0.6736** | **0.7404** | **37** | **0.3693** | **0.4926** | **51** | **0.4071** | **0.4633** | **48** |

Table 9: Zero-Shot Evaluation on HotpotQA of Models Trained on Natural Questions.

|  | EM | F1 | #tok |
|---|---|---|---|
| No Retrieval | 21.05 | 29.48 | 0 |
| Full Documents | 32.99 | 43.69 | 737 |
| BM25 | 24.18 | 35.73 | 71 |
| NoiseFilter-IB | 27.97 | 38.62 | 38 |
| RECOMP | 28.96 | 39.95 | 56 |
| CORE | 33.67 | 45.06 | 36 |
| **RECOMP-Transfer** | **26.68** | **37.29** | **58** |
| **CORE-Transfer** | **31.25** | **42.84** | **35** |

Table 10: Evaluation on Noisy Natural Questions.

|  | EM | F1 | #tok |
|---|---|---|---|
| full documents | 35.21 | 45.38 | 1427 |
| RECOMP | 33.29 | 43.90 | 59 |
| **CORE** | **38.19** | **48.85** | **48** |

# E    CROSS-DATASET GENERALIZATION PERFORMANCE

To verify the universal compression capability of our method, we directly transfer the model trained on the single-hop question answering dataset NQ to the multi-hop dataset HotpotQA for evaluation. The results, shown in Table 9, indicate that our transferred model achieves nearly lossless performance compared to using full documents without compression, while substantially outperforming the transfer results of the RECOMP baseline. Moreover, although both our method and the baseline underperform relative to models trained directly on the target HotpotQA dataset, our approach exhibits a smaller performance drop and demonstrates greater robustness compared to the baseline.

# F    ROBUSTNESS AGAINST NOISY CONTEXTS

To evaluate the robustness of our approach against adversarial retrievals and noisy contexts, we constructed a noisy version of the NQ dataset. For each question, we constructed the input context by combining the top-3 passages retrieved by the DPR retriever with 7 randomly selected passages from the Wikipedia corpus to serve as irrelevant/noisy information. This resulted in a context of 10 passages, which were then shuffled to randomize the order. We then compared the performance of our method against the full-document baseline. Experimental results are presented in the table 10. In the "full documents" setting, the downstream LLM directly uses all these 10 passages to answer the question, whereas in our method, the compressor first summarizes the context, and the LLM then generates an answer based on the compressed content. The model we used was trained on the standard NQ dataset without any such noise augmentation. Our method not only matches but slightly surpasses the performance of using all documents, demonstrating its strong noise resistance and ability to extract key information from cluttered contexts. In addition, we compared our approach with the RECOMP baseline, and our method consistently outperforms it, reaffirming the superior compression capability and robustness of our model. Furthermore, our method achieves a high compression rate, condensing the source content from 1,427 tokens to just 48.

