# OpenReview forum: "CORE: Lossless Compression for Retrieval-Augmented LLMs via Reinforcement Learning"
_ICLR.cc/2026/Conference — Submitted to ICLR 2026_

### Official Review · Reviewer_XUrq · 2025-10-21

**Soundness:** 3
**Presentation:** 4
**Contribution:** 2
**Rating:** 4
**Confidence:** 4

**Summary:**

This paper proposes CORE, a method that trains a compressor model that compresses long evidence documents into concise summaries, using end-to-end reinforcement learning that directly optimizes downstream task performances. Experiments adopted four QA tasks and showed that the compressed evidence achieves better or on-par performances with using the original evidence on different downstream models, with a high compression rate (~95% reduction in tokens). Ablations also suggest that CORE may transfer across compressor model families and downstream model families.

**Strengths:**

This paper is well written with clear presentation and logical comprehensiveness. It proposes a strong engineering implementation of an end-to-end RL framework (and models) for evidence compression, which achieved strong empirical results on multiple common QA datasets with a very high compression rate. The experiments contained multiple baselines for future work to reference. The provided implementation details support reproducibility.

**Weaknesses:**

My main concern with this paper is the academic contribution to the ICLR community. While CORE presents a strong engineering implementation that may benefit future research and applications, the general idea that uses RL for compression optimization can be found in papers from years ago, such as  [1], and some newer works from months ago use task performances as rewards, such as [2] and [3]. The authors need to better discuss key differences between these works (other than better engineering implementation), and the current related works section is insufficient in my view. Moreover, the authors should consider some e2e RL baselines in experiments.

At the same time, using task performances as rewards itself is somewhat concerning from a high level, because the compressor model is leaked with task labels, and will optimize towards "solving the problem", rather than "being faithful to the evidence". In other words, it seems to me that the compressor model will tend to solve the problem itself, and may limit task transfer performances on more challenging reasoning tasks (which I suggest the authors experiment with) rather than direct IR tasks such as those evaluated in the paper.

Some misleading terms exist, such as "loselss," which typically refers to information lost from the original evidence instead of performance losses. In order to claim lossless, the authors should conduct studies to show that key information related to the (comprehensive and unbiased, not just performance-oriented) reasoning of the problem is preserved.


[1] Discrete Prompt Compression with Reinforcement Learning, Jung and Kim, 2023
[2] Oreo: A Plug-in Context Reconstructor to Enhance Retrieval-Augmented Generation, Li and Ramakrishnan, 2025
[3] TACO-RL: Task Aware Prompt Compression Optimization with Reinforcement Learning, Shandilya et al. 2024

**Questions:**

Please refer to the weakness section. I do not have particular confusions about the current manuscript as it is well written.

---

> ### Author Response · Authors · 2025-11-21
> **Response to Reviewer XUrq [Part I]**
>
> **Dear Reviewer XUrq**:
>
> We sincerely appreciate your insightful comments and your kind recognition of our work. In the following, we provide a detailed response to each of the points you have raised.
>
> ***`Response to W1: Differentiation and Contribution`***
>
> We thank the reviewer for highlighting the three related papers [1,2,3]. The key distinctions between our approach and these cited studies are detailed in the following. *Additionally, **a condensed discussion of these works has been incorporated into the Related Work section of our revised paper and highlighted in red** for your convenience.*
>
> **The key distinctions from References [1] and [3] are outlined as follows:**
>
> - **Object of Compression: Prompt vs. Documents in RAG.** Both PCRL [1] and TACO-RL [3] are designed for prompt compression. The fundamental focus of our work (CORE) is different: we concentrate on compressing documents in a multi-document RAG setting. We observe that existing compression methods in RAG are largely heuristic and often lead to degraded performance. To address this, we propose a lossless compression strategy based on end-task effectiveness. This task itself presents greater challenges, which we aim to tackle with our method.
>
> - **Compression Method: Discriminative/Extractive vs. Generative/Abstractive.** Both PCRL[1] and TACO-RL [3] perform discriminative (or extractive) compression. Their action space is binary: to keep or drop each token from the original prompt. The output is a strict subset of the input. Different from them, CORE performs generative (or abstractive) compression. Our compressor can rephrase, condense, and synthesize information, generating novel text that may not appear verbatim in the source documents. This provides greater flexibility and potential for higher compression rates, which are validated by our experiments.
>
> - **Fundamental RL Formulation and Reward Signal.** Both PCRL [1] and TACO-RL [3] employ a reward signal based on the similarity between the outputs of the LLM when fed with uncompressed versus compressed prompts. PCRL [1] formulates the compression policy optimization through a Markov decision process combined with a contextual multi-armed bandit, while TACO-RL [3] adopts the classical on-policy REINFORCE algorithm. In contrast, CORE introduces a direct task-performance reward, computed by comparing the task label with the output of the end-task LLM. This reward is optimized using the GRPO algorithm, which not only enhances training stability and sample efficiency but also effectively aligns the compressed model with downstream task objectives.
>
> - **Methodological Innovation.** To provide a robust initial policy for reinforcement learning (RL) and ensure training stability, we employ knowledge distillation from a teacher model to initialize our compressor. This approach establishes a strong foundation for the subsequent RL phase, which utilizes GRPO. Our ablation studies confirm the indispensable contribution of this initialization stage—without it, lossless performance cannot be achieved. This design constitutes another key differentiator from the related works mentioned by the reviewers, namely PCRL [1] and TACO-RL [3].
>
> - **Performance and Compression Ratio.** While both [1] and [3] degrade end-task performance, our approach achieves lossless compression with a significantly higher ratio. Specifically, PCRL [1] reports a 3% performance drop at a modest 24.6% compression rate. TACO-RL [3] suffers a 2-point performance decrease even at 50% compression, which plummets to over 10 points at a compression ratio of 1/6. In contrast, our method, CORE, not only preserves performance but improves the average EM score on downstream tasks by 3.3 points. Moreover, it accomplishes this at an aggressive compression rate of 3%, demonstrating true lossless compression without any performance penalty.
>
> ***Due to space limitations, please refer to Part II for our continual response to W1.***

---

> ### Author Response · Authors · 2025-11-21
> **Response to Reviewer XUrq [Part II]**
>
> ***`Response to W1: Differentiation and Contribution (continue1)`***
>
> **The key distinctions from References [2] are outlined as follows:**
>
> - **Different Motivations and Objectives.** Oreo aims to enhance RAG performance by reconstructing and refining retrieved chunks. In contrast, our method, CORE, is primarily designed to compress the content of retrieved documents to improve inference efficiency, while ensuring no degradation in performance. Although our compression process also results in some performance gains, this is considered a secondary benefit. This distinction is also reflected in the experimental evaluations: Oreo's analysis focuses mainly on performance metrics, whereas our paper, CORE, provides a comparative analysis of both performance and compression ratios (Table 1).
>
>
> - **Different Policy Optimization.** Oreo employs PPO for policy optimization, which relies on a value function that can introduce estimation bias and instability. In contrast, CORE adopts GRPO, which leverages a group-wise comparison approach to eliminate the need for a value function, removing its estimation bias and instability while simplifying the algorithm architecture. This results in more efficient, robust, and stable policy optimization.
>
>
> - **Different Reward Designs.** Oreo utilizes a sparse, sequence-level ROUGE score, which is then distributed across tokens using a weighting mechanism based on the policy's confidence. Note that the indirect ROUGE-based reward is a mere proxy for task performance. In contrast, our method CORE employs simpler, rule-based rewards directly derived from end-task QA performance. We combine a binary Exact Match (EM) reward with a finer-grained F1 score that credits partial correctness. This results in a denser, more instructive learning signal compared to the baseline's sparse, probabilistically-weighted reward.
>
>
> - **Different Method Designs.**  Oreo employs a complex three-stage training paradigm. Its contrastive learning phase requires the construction of a large number of contrastive samples, and the difficulty in controlling the quality of these samples introduces additional training uncertainty. In contrast, our approach adopts a simpler and more straightforward two-stage paradigm. Furthermore, for efficiency reasons, our compressor is intentionally designed to be significantly smaller than the downstream LLM to reduce the computational cost of encoding retrieved documents. Oreo, however, does not take this into account. It uses T5-small (0.06B) as the compressor and Flan-T5 (0.2B) as the downstream model—the size difference between them is only about 3×, and both models are relatively small in scale, not qualifying as large language models. In comparison, our method uses Qwen2.5 -1.5B as the compressor and Qwen2.5 -14B as the downstream model. The compressor has only about **1/10** the parameters of the downstream model. Moreover, the model scales we use are considerably larger than those in Oreo, which better demonstrates the generalization capability of our approach.
>
>
> Moreover, we are greatly encouraged that the **other three reviewers have all positively acknowledged the novelty of our work**. Reviewer pJpM commented that "The core idea is innovative", Reviewer HZQP noted that "The paper introduces an innovative approach", and Reviewer LyaC highlighted that it "provides a good empirical and conceptual motivation". We believe these assessments further reinforce the unique contribution of our method in comparison to existing literature.
>
> Reference:
>
> [1] Discrete Prompt Compression with Reinforcement Learning, Jung and Kim, 2023
>
> [2] Oreo: A Plug-in Context Reconstructor to Enhance Retrieval-Augmented Generation, Li and Ramakrishnan, 2025
>
> [3] TACO-RL: Task Aware Prompt Compression Optimization with Reinforcement Learning, Shandilya et al. 2024
>
>
> ***Due to space limitations, please refer to Part III for our continued response to W1.***

---

> ### Author Response · Authors · 2025-11-21
> **Response to Reviewer XUrq [Part III]**
>
> ***`Response to W1: Differentiation and Contribution (continue2)`***
>
> **Regarding *“the authors should consider some e2e RL baselines in experiments.”***
>
> - We note the reviewer’s mention of end-to-end (e2e) reinforcement learning baselines. We would like to clarify that e2e RL methods such as ReSearch [5] and R1-searcher [6] **fall outside the scope of our problem setting**, although they are discussed in our related work section for comprehensiveness. As clearly stated in Section 2.1 (Problem Formulation) of our paper: ***“We treat the generator LLM as a black-box system and focus exclusively on training the compressor model.”*** This means the parameters of the generator LLM are unknown and remain fixed throughout the training process.
>
> - It is worth noting that **this setting follows our precursor work**, RECOMP [4].
>
> - In contrast, recent e2e RL methods like ReSearch and R1-searcher use RL to directly train the generator LLM—meaning its parameters are updated during training, which violates the black-box assumption. Therefore, these methods are not within our comparison framework. Moreover, it is important to emphasize that generator LLMs are typically very large, and directly training them incurs significantly higher costs compared to training a smaller compressor model. In addition, many LLMs do not offer open access to their parameters. These practical constraints further validate the relevance and motivation of our problem setting.
>
> Reference
>
> [4] Recomp: Improving retrieval-augmented lms with context compression and selective augmentation, Xu et al. 2024
>
> [5] Learning to reason with search for llms via reinforcement learning, Chen et al. 2025
>
> [6] R1-searcher: Incentivizing the search capability in llms via reinforcement learning, Song et al. 2025

---

> ### Author Response · Authors · 2025-11-21
> **Response to Reviewer XUrq [Part IV]**
>
> ***`Response to W2: concerns about using task performances as rewards and task transfer ability`***
>
> We thank the reviewer for the comment. We address the reviewer's concerns from the following aspects:
>
> - We clarify that there is **no leakage of task labels** into the compressor. The compressor's input is only the question and retrieved documents. The task labels (answers) are used solely as an RL reward signal to score the compressor's output, which is a standard supervised learning setup, not leakage.
>
> - **Our reward does not optimize the compression model toward "solving the problem"**, because it is the downstream generator LLM—not the compressor LLM—that is responsible for producing the final predicted answer. The compressor LLM only generates a compressed summary, which is then combined with the original question and fed into the downstream generator LLM to produce the predicted answer. The reward is computed based on this predicted answer and the gold answer. **Therefore, our reward optimizes the compression model to assist the downstream generator LLM in solving the problem, rather than optimizing the compression model to solve the problem itself**.
>
> - The RAG framework is typically applied to knowledge-intensive tasks, with multi-document question answering being the most representative scenario. Our experimental evaluation covers two single-hop and two multi-hop QA datasets. For the multi-hop reasoning tasks (HotpotQA, 2WikiMultihopQA), where the generator LLM must perform complex reasoning to get the answer, using documents compressed by our method results in better downstream performance compared to using the original documents (see Table 1). This clearly demonstrates that our approach is suitable for reasoning tasks and achieves strong performance.
>
> - **Task Transfer Performance.** To better verify our method’s task transfer capability, **we have conducted an additional transfer experiment**: using a model trained on one dataset (NQ) and evaluating it on another dataset (HotpotQA). Note that HotpotQA needs the LLM to perform complex reasoning to get the answer. The results are presented below. ***In addition, the experiments are also presented in the revised paper (see Appendix E and Table 9)***.
>
> |  | EM | F1 | #tok |
> |---|:---:|:---:|:---:|
> | No Retrieval | 21.05 | 29.48 | 0 |
> | Full Documents | 32.99 | 43.69 | 737 |
> | BM25 | 24.18 | 35.73 | 71 |
> | NoiseFilter-IB | 27.97 | 38.62 | 38 |
> | RECOMP | 28.96 | 39.95 | 56 |
> | CORE | 33.67 | 45.06 | 36 |
> | **_RECOMP-Transfer_** | **_26.68_** | **_37.29_** | **_58_** |
> | **_CORE-Transfer_** | **_31.25_** | **_42.84_** | **_35_** |
>
> - The results indicate that our transferred model achieves nearly lossless performance compared to using full documents without compression, while substantially outperforming the transfer results of the RECOMP baseline. Moreover, although both our method and the baseline underperform relative to models trained directly on the target HotpotQA dataset, our approach exhibits a smaller performance drop and demonstrates greater robustness compared to the baseline. **The experimental results provide strong evidence for the task-transfer capability of our method**.
>
> ***`Response to W3: the term of “lossless”`***
>
> - We thank the reviewer for this insightful comment. **We agree that the term "lossless" can be ambiguous in this context**. We apologize for any confusion caused by our wording.
>
> - **In our work, we use "lossless" specifically to mean "performance-lossless" or "task-lossless"**. Our key claim is that the compressed content produced by our method does not lead to any degradation in the downstream LLM's performance on its primary task compared to using the original, uncompressed context. We have conducted extensive experiments (as shown in Table 1 and Table 2 of our paper) to validate that the performance metrics (EM and F1-score) remain statistically equivalent or even improve.
>
> - We acknowledge that this does not constitute a formal proof of information-theoretic losslessness, as some original textual details are inevitably removed. Our focus is on preserving the information necessary for accurate reasoning, rather than information-theoretic losslessness. **To avoid misunderstanding, we have revised the title of our paper by replacing "lossless compression" with the more precise term "performance-preserving compression."** Specifically, the new title now reads: "CORE: Performance-Preserving Compression for Retrieval-Augmented Generation."

---

> > ### Comment · Reviewer_XUrq · 2025-11-25
> >
> > I thank the authors for their rebuttal. Regarding the comparisons between the related approaches I mentioned, I think they are currently insufficient to justify what the paper proposes. For example, prompt compression sounds different, but it can also be applied directly to the evidence compression task. In essence, the authors need to justify *why* we need the proposed CORE method, and what kind of key problems CORE addresses but previous works conceptually (or empirically) cannot. In other words, just being "new" is not enough to claim contribution (especially since I do not think it's very new, conceptually). I would appreciate it if the authors could provide a concise explanation of what kind of evidence suggests that I need CORE.
> >
> > Regarding the data leakage, I am suggesting that the compressor somewhat has knowledge regarding the task label when generating compressions, and especially with RL, it seems to be inevitable that a capable compressor will start to try to solve the problem, as a way to generate better compressions towards a better task performance. What kind of evidence can the authors provide that the compressor is still faithful to the "compression" task?

---

> > > ### Author Response · Authors · 2025-11-26
> > > **Response to comparisons between the related approaches**
> > >
> > > Thanks for the reviewer's feedback.
> > >
> > > Regarding your first concern, let us concisely explain ***“why you need CORE.”***
> > >
> > > ***`That is, “if you want to achieve performance-lossless document compression with a significantly high compression ratio for RAG, you need CORE”.`*** **This is a key problem that CORE addresses but prior works empirically (and conceptually) fail to solve**.
> > >
> > > We will provide a detailed explanation from both **empirical and conceptual** perspectives.
> > >
> > > **Empirically**:
> > > - **Performance and Compression Ratio**: As for the prompt compression papers you mentioned, their methods lead to a performance **drop of 2 to 3 points** compared to using uncompressed data—an unacceptable outcome for tasks requiring high precision. In contrast, CORE achieves performance-lossless compression, not only preserving original performance but even **improving it by 1–2 points** in general. Moreover, their approaches are specifically designed for prompt compression; even if generalized to RAG, their effectiveness would further decline.
> > > As for the compression ratio, their method only achieves a compression ratio of **20% to 50%**, whereas **CORE reaches 3%**, representing a substantial improvement in compression efficiency.
> > >
> > >
> > > **Conceptually:**
> > >
> > > - **Conceptual Explanation for Superior Performance**
> > >   - **Task-Oriented Reward Mechanism**: Those prompt compression methods used sentence-level structural similarity (between the compressed and original prompts) as a reward signal. Crucially, this is not an end-task metric, so it cannot guarantee that the compressed content remains optimal for the final application. In contrast, our approach directly uses the end-task performance metric as the reward, ensuring the compression is explicitly guided to be beneficial for the downstream task.
> > >   -  **Distillation-Based Warm-Start**: We introduce a knowledge distillation phase prior to RL training. This serves as a critical warm start, providing the RL policy with a high-quality initial point from which to optimize. Our ablation studies confirm that this step is vital for achieving peak performance—a conceptual element entirely absent in prior works that the reviewer mentioned.
> > >   -  **Advanced Optimization with GRPO**: We optimize our compressor using the GRPO algorithm with rule-based reward signals. This approach provides distinct advantages, most notably by replacing PPO's difficult-to-train reward model. GRPO employs a group-based relative reward baseline, which effectively reduces the variance of gradient estimates and leads to more stable and efficient training.
> > >
> > >
> > >
> > >
> > > - **Conceptual Explanation for Higher Compression Ratio**
> > >    - Previous methods typically rely on token selection for compression. Their action space is binary: to keep or drop each token from the original text. This approach is fundamentally limited because it cannot reorganize information; it merely filters the original, often redundant, content. Even after selecting important tokens, the output remains a subset of the original sequence, inheriting its inherent redundancies. It's like creating a shorter version of a book by removing some sentences—the fundamental narrative flow and potential wordiness remain.
> > >    - In contrast, our method employs a generative summarization approach. It operates in a much larger action space: it can paraphrase, condense concepts, and synthesize new sentences that capture the semantic essence without the lexical baggage. This is analogous to writing a summary of the book in your own words. You are not bound by the author's original phrasing, allowing you to express the core ideas far more efficiently.
> > >
> > >
> > > **Conclusion:**
> > >
> > > **Given these comparisons—if your goal is performance-lossless compression with a high compression ratio for RAG—which method would you choose?**
> > >
> > > ***`Clearly, the answer is CORE,`*** as it is the only approach capable of achieving this.

---

> ### Author Response · Authors · 2025-11-26
> **Response to the second concern "faithful to the compression task"**
>
> Regarding the second concern raised by the reviewer about whether the compressor remains faithful to the compression task itself, we address this issue from the following perspectives:
>
> **First, we outline the efforts our method has made to ensure that the compressor remains faithful to the compression task.**
>
> - **Distillation Learning Phase**: Prior to reinforcement learning training, we incorporated a distillation learning phase. During this stage, the compression model is trained via supervised fine-tuning (SFT) to learn how larger models (DeepSeek) summarize documents, rather than answering the question directly. This is achieved by **using *summary of DeepSeek* as the training label** instead of *the answer to the question*.
>
> - **Prompt Template**: The instruction in our prompt template explicitly requires the compressor to output a summary rather than an answer. Our prompt is as follows:
> ***"Compress the information in the retrieved documents into a 2-sentence summary that could be used to answer the question. If the documents do not contain relevant information, simply output an empty string."***
> All compressors used in our study are fine-tuned from instruction-tuned LLMs, which are expected to possess strong instruction-following capabilities.
>
> - **Case Study**: We have included a case study in Appendix C, which presents examples of the content generated by the compressor. As shown, the output consists of summarized text rather than direct answers to the question.
>
> **Second**, regarding *"it seems to be inevitable that a capable compressor will start to try to solve the problem, as a way to generate better compressions towards a better task performance."* We provide the following explanation.
> -  The compressor has a significantly smaller number of parameters than the downstram LLM—only about **1/10** of those in the downstream LLM. As a result, its knowledge capacity is significantly more limited. Given this constraint, the compressor’s ability to answer questions directly is considerably weaker than that of the downstream LLM. Therefore, using the compressor to answer questions directly would not lead to better performance. In contrast, our method achieves superior experimental results compared to using the downstream LLM alone, which indirectly confirms that the compressor provides additional informative input to the downstream LLM rather than attempting to answer the question directly.
>
>
> **Finally, and most importantly, we would like to emphasize that the primary goal of this paper is to generate compression that preserves/enhances downstream performance, not information-theoretic compression.** While faithfulness is a good attribute, it does not necessarily lead to improved final outcomes. In other words, **a compression that is optimal in a general or information-theoretic sense may not be the most useful for a specific downstream task**.
>
> This work focuses precisely on pursuing the form of compression that is most beneficial to the downstream task. That is to say, the compression we pursue is not traditional information-theoretic compression. It can involve rewriting and reconstructing the original text, or generating any content based on the source document that proves useful for the downstream task—all such approaches are welcome as long as they preserve/enhance downstream performance. Moreover, our experiments validates that our compressor can achieve this goal.

---

### Official Review · Reviewer_LyaC · 2025-10-29

**Soundness:** 3
**Presentation:** 3
**Contribution:** 2
**Rating:** 6
**Confidence:** 4

**Summary:**

The paper proposes CORE which is an end-to-end framework for context compression in RAG for LLMs using RL. CORE uses downstream task performance as direct reward when training the compressor. The method is evaluated on four QA datasets with EM and F1 as the evaluation metrics, with RECOMP and NoiseFilter as baselines.

**Strengths:**

- The paper focuses on the challenges of context explosion in RAG paradigms, providing a good empirical and conceptual motivation.
- The ablation studies on document # and cross backbone LLMs are comprehensive and demonstrate effectiveness.
- The paper is mostly clear in the presentation and has an anonymous code repository.

**Weaknesses:**

- Missing experiments with strong compressor baselines and datasets on context compression. The paper mostly tested on QA datasets with short retrieved document length. It would be neccessary to experiment on QA benchmark with long document context, including ZeroScrolls and LongBench. Also compare with baselines including LLMLingua-2, LongLLMLingua.
- Could the authors clarify whether their approach is robust against adversarial retrievals or irrelevant/noisy context (e.g., randomly shuffled or negative docuements)?

**Questions:**

See the weakness.

---

> ### Author Response · Authors · 2025-11-21
> **Response to Reviewer LyaC [Part I]**
>
> **Dear Reviewer LyaC**:
>
> We sincerely appreciate your insightful comments and your kind recognition of our work. In the following, we provide a detailed response to each of the points you have raised.
>
> ***`Respond to W1: baselines and datasets`***
>
> We sincerely thank the reviewer for the constructive suggestion.
>
> **Baselines**:
>
> - We thank the reviewer for the suggestion regarding baseline comparisons. In response, we have incorporated two strong compressor baselines—LongLLMLingua [1] and QGC [2]—into our evaluation. LongLLMLingua is a method specifically designed for long-text compression, while QGC is a query-guided compression approach.
>
> - As shown in the below table, QGC outperforms LongLLMLingua in terms of both performance and compression rate. Nevertheless, our proposed method, CORE, achieves the best results across all evaluation metrics, attaining the highest accuracy with the lowest compression rate. These supplementary experiments further validate the effectiveness of our approach. The corresponding results have been added to Table 1 in the revised paper and are highlighted in red for ease of reference.
>
> |  | NQ |  |  | TriviaQA |  |  | HotpotQA |  |  | 2Wiki |  |  |
> |---|---|---|---|---|---|---|---|---|---|---|---|---|
> |  | EM | F1 | #tok | EM | F1 | #tok | EM | F1 | #tok | EM | F1 | #tok |
> | LongLLMLingua  | 33.65 | 43.15 | 152 | 58.96 | 66.82 | 148 | 28.03 | 38.49 | 149 | 29.37 | 33.62 | 153 |
> | QGC | 36.23 | 45.88 | 49 | 61.02 | 68.45 | 47 | 29.16 | 40.05 | 45 | 31.14 | 36.83 | 51 |
> | **CORE** | **41.02** | **50.4** | **46** | **65.63** | **72.55** | **32** | **33.67** | **45.06** | **36** | **36.72** | **42.05** | **49** |
>
>
> [1] Longllmlingua: Accelerating and enhancing llms in long context scenarios via prompt compression, ACL 2024
> [2] Retaining key information under high compression ratios: Query-guided compressor for llms, ACL 2024
>
>
> **Datasets**:
>
> - We thank the reviewer for pointing out the LongBench and ZeroSCROLLS benchmarks. The key distinction lies in their nature as general-purpose, multi-task benchmarks covering diverse problems like summarization and code completion. In contrast, our work specifically targets multi-document question answering task, which is a core and predominant task in RAG systems. To ensure a focused and fair comparison, our dataset selection and experimental setting are **aligned with established baseline methods**, including RECOMP [3], QGC [2], and NoiseFilter-IB [4]. Following these related works, we evaluate our method on four multi-document question-answering datasets: NQ, TriviaQA, HotpotQA, and 2WikiMultihopQA. For each dataset, we use a context composed of 10 documents retrieved by the retrieval model. Specifically, the total length of these 10 documents reaches 1.5k tokens, which is twice the maximum context length used in RECOMP (limited to five documents), and matches the context length adopted in QGC.
>
> - In addition, in preliminary experiments, we also investigated the effect of increasing the number of retrieved documents on the original RAG performance. As shown in Figure 1 of the original paper, we observed diminishing marginal gains beyond 10 documents, which motivated our choice to evaluate across document counts ranging from 1 to 10. Furthermore, we conducted generalization tests on input context length, demonstrating that our method can handle significantly longer documents during inference than those seen during training. Detailed results are provided in Table 1 of the original paper under the experiment titled “Compression of top 10 documents (with the compressor trained on top 5 docs).” These experimental setups collectively affirm that our evaluation adequately addresses the scenario of long-document contexts, as raised by the reviewers.
>
> [3] Recomp: Improving retrieval-augmented lms with context compression and selective augmentation. ICLR 24
>
> [4] An information bottleneck perspective for effective noise filtering on retrieval-augmented generation. ACL 2024
>
> ***Due to space limitations, please refer to Part II for our responses to other questions.***

---

> ### Author Response · Authors · 2025-11-21
> **Response to Reviewer LyaC [Part II]**
>
> ***`Respond to W2: whether the approach is robust against irrelevant/noisy context `***
>
> - We sincerely thank the reviewer for the valuable suggestion. To evaluate the robustness of our approach against adversarial retrievals and noisy contexts, we constructed a noisy version of the NQ dataset. For each question, we constructed the input context by combining the top-3 passages retrieved by the DPR retriever with 7 randomly selected passages from the Wikipedia corpus to serve as irrelevant/noisy information. This resulted in a context of 10 passages, which were then shuffled to randomize the order. We then compared the performance of our method against the full-document baseline.
>
> - Experimental results are presented in the table below.
> |  | **NQ (noisy context)** |  |  |
> |:---:|:---:|:---:|:---:|
> |  | EM | F1 | #tok |
> | full documents | 35.21 | 45.38 | 1427 |
> | RECOMP | 33.29 | 43.90 | 59 |
> | **CORE** | **38.19** | **48.85** | **48** |
>
> - In the “full documents” setting, the downstream LLM directly uses all these 10 passages to answer the question, whereas in our method, the compressor first summarizes the context, and the LLM then generates an answer based on the compressed content. The model we used was trained on the standard NQ dataset without any such noise augmentation.
>
> - **Our method not only matches but slightly surpasses the performance of using full documents**, demonstrating its strong noise resistance and ability to extract key information from cluttered contexts. In addition, we compared our approach with the RECOMP baseline, and our method consistently outperforms it, reaffirming the superior compression capability and robustness of our model. Furthermore, our method achieves a high compression rate, condensing the source content from 1,427 tokens to just 48.
>
> - In accordance with your suggestion, the relevant analysis has been incorporated into the revised paper as Appendix F and Table 10.

---

> ### Comment · Reviewer_LyaC · 2025-11-26
>
> Thanks for the experiments with SOTA compressors and nosiy doc ablation. I still have concern that the benchmarks tested in the paper are not standard for long-context compressors. LongBench and ZeroSCROLLS contain tasks such as multi-doc QA and query-based summarization, which falls in the scenario discussed in this paper.

---

> > ### Author Response · Authors · 2025-11-27
> > **Addressing Benchmark Concerns**
> >
> > **Dear Reviewer LyaC**,
> >
> > We sincerely thank you for your timely feedback regarding the benchmarks.
> >
> > We are grateful for your suggestion on the **LongBench**'s multi-doc QA task, which will help us enhance the experimental section of our paper.  Following this suggestion, **we have incorporated an evaluation using LongBench to further validate the effectiveness of our proposed method**.
> >
> > LongBench includes three English multi-doc QA datasets: MuSiQue, HotpotQA, and 2WikiMultihopQA. The average document lengths for these datasets are 11,214, 9,151, and 4,887 tokens, respectively, making them representative for long-context scenarios.
> >
> > We compared four settings on these three LongBench datasets:
> >
> > 1. No document context (i.e., the input contains only the question),
> > 2. Using the full documents without compression,
> > 3. Compressing documents using LongLLMLingua,
> > 4. Compressing documents using our method, CORE.
> >
> > In all settings, we employed Qwen2.5-14B as the LLM for answer generation. For the trainable compression methods, both were fine-tuned using Qwen2.5-1.5B-Instruct on the HotpotQA training set that we originally used (10 docs) in the paper.
> >
> >
> > The experimental results, including performance and compression ratios, are presented in the table below.
> >
> > - ***Table: Comparison Results on Multi-Doc QA Tasks of LongBench***
> >
> > | **LongBench** |        MuSiQue |  |        HotpotQA |  |        2WikiMQA |  | Average |  |
> > |---|:---:|:---:|:---:|:---:|:---:|:---:|:---:|:---:|
> > |  | _F1 score_ | _length_ | _F1 score_ | _length_ | _F1 score_ | _length_ | _F1 score_ | _length_ |
> > | no documents | 26.49 | 0 | 51.13 | 0 | 43.95 | 0 | 40.52 | 0 |
> > | full documents (without compression) | 38.41 | 11214 | 62.22 | 9151 | 57.97 | 4887 | 52.87 | 8417 |
> > |     LongLLMLingua | 33.18 | 983 | 57.69 | 907 | 53.26 | 489 | 48.04 | 793 |
> > | **CORE** | **38.94** | **129** | **63.58** | **126** | **59.73** | **108** | **54.08** | **121** |
> >
> >
> > As can be observed, using full documents leads to a significant improvement in performance compared to using no relevant documents as context, though it also substantially increases the context length. As for compression methods, when LongLLMLingua is applied to compress the documents, although the document length is reduced to some extent, the F1 score also decreases relative to the performance achieved with the original full documents—by an average of 4 to 5 points. In contrast, **when our compression method CORE is applied, not only is an extremely high compression rate of about 2% achieved, but the performance remains lossless across all datasets, with even a slight improvement observed in all cases.** The experimental results on LongBench further validate the effectiveness of our method in long-context compression scenarios.
> >
> >
> > *(Note: The HotpotQA and 2WikiMultihopQA used in LongBench differ from those used in Table 1 of our paper. The primary distinctions lie in the context length and the methodology for document retrieval and construction.)*
> >
> > **We greatly appreciate your insights and look forward to your further feedback.**

---

### Official Review · Reviewer_HZQP · 2025-10-31

**Soundness:** 2
**Presentation:** 3
**Contribution:** 2
**Rating:** 4
**Confidence:** 4

**Summary:**

The paper proposes CORE, a novel method for lossless context compression in Retrieval-Augmented Generation (RAG) models. It aims to address the inefficiencies in current RAG systems, where increasing the number of retrieved documents leads to a higher computational cost without necessarily improving the model's performance. CORE leverages reinforcement learning to optimize a compression policy that improves task performance without requiring predefined compression labels. The method uses downstream task performance as feedback, iteratively refining the compression strategy to prevent performance degradation. The authors demonstrate its effectiveness with experimental results across four benchmark datasets.

**Strengths:**

1. The paper introduces an innovative approach by combining reinforcement learning with end-to-end optimization to address the lossless compression problem in RAG models. This method is distinct from previous heuristic-based approaches that fail to align compression with downstream task performance.

2. The use of Group Relative Policy Optimization (GRPO) for end-to-end training of the compressor without requiring predefined summaries or labels is a clear strength. This approach ensures the compression is aligned directly with the task's requirements.

3. The paper claims that CORE not only prevents performance degradation but also improves task performance (up to 3.3 EM points) with significantly reduced computational costs. The method demonstrates strong generalization across different datasets and model architectures. The ablation study shows that both the distillation warm-up and GRPO phases are critical, and the results remain consistent across multiple architectures (e.g., LLaMA, Qwen).

**Weaknesses:**

1. The method's computational efficiency is presented in terms of compression ratios, but there is little discussion of how the approach scales when both the retrieved documents and the LLMs are larger. For example, the computational overhead of training the compressor and fine-tuning on new tasks could be prohibitive for very large-scale deployment.

2. While the results are promising, the paper lacks a detailed discussion of the interpretability of the compressed summaries generated by CORE. In real-world applications, understanding why certain documents or parts of documents are prioritized for compression could help improve trust in the system. The case studies presented (on NQ and 2Wiki datasets) are useful, but further case studies or qualitative analysis of failure cases could strengthen the argument for the method's robustness.

3. The paper mostly compares CORE with a few existing compression methods (e.g., RECOMP, NoiseFilter-IB) and traditional RAG. While these are important baselines, the method could be better contextualized within the broader field by incorporating more recent compression techniques or exploring how CORE performs against other forms of task-specific compression like query-guided methods.

**Questions:**

1. Could you provide additional insights into how CORE scales with the size of both the LLM and the number of retrieved documents? How would it perform in environments where the retrieved document pool and LLM models are very large?

2. What is the computational cost of training the CORE model compared to traditional RAG methods? Specifically, how does the fine-tuning process scale when switching between different LLMs?

3. How can we better understand the decisions made by the compressor during the document selection and compression process? Is there a way to visualize or explain why specific content is prioritized?

4. Does CORE have any potential applicability in non-textual domains (e.g., image captioning, multimodal RAG systems)?

---

> ### Author Response · Authors · 2025-11-21
> **Response to Reviewer HZQP [Part I]**
>
> **Dear Reviewer HZQP**:
>
> We sincerely appreciate your insightful comments and your kind recognition of our work. In the following, we provide a detailed response to each of the points you have raised.
>
> ***`Response to W1 & Q1: how the approach scales when both the retrieved documents and the LLMs are larger`***
>
> Thanks for the reviewer's comment about the computational cost scaling. We address the question from the following three perspectives.
>
> - **Scaling with the number of retrieved documents:** As the number of retrieved documents increases, the computational overhead rises across all methods, primarily due to the longer input sequences processed by the compressor model, which extends its encoding time. However, compared to other RAG methods, the efficiency gains achieved by our approach become more pronounced as more documents are retrieved. This is because our method uses a smaller compression model instead of a large generator LLM to handle the lengthy text encoding. Moreover, it is important to note that when the number of documents grows, our method generalizes effectively from fewer to more documents without requiring retraining. As shown in Table 1 and discussed in the last paragraph of Section 3.2, compressors trained on 5 documents maintain high performance and compression rates when applied to 10 documents without any retraining. This significantly reduces the cost associated with repeatedly retraining the compressor.
>
>
> - **Scaling with compressor model size:** Increasing the size of the compressor model leads to higher computational overhead during both training and inference, as more parameters result in longer forward and backward propagation time. To mitigate this, our approach incorporates two key designs. First, the compressor is lightweight by design—typically an order of magnitude smaller than the generator LLM—while remaining lossless, thus avoiding the need for a large model size. Second, only the lightweight compressor is updated during training, while the larger generator LLM remains fixed, which substantially reduces training overhead. Moreover, as demonstrated in Section 3.3, increasing the compressor size can improve performance, presenting a clear trade-off between model capacity and computational cost. Therefore, the  compressor model size can thus be selected according to specific requirements for performance and computational budget.
>
>
> - **Scaling with the size of the generator LLM:** During training, the generator LLM remains fixed and is accessed only via API to generate responses for reward computation. Therefore, increases in its size have a minimal impact on training overhead. During inference, compared to an uncompressed baseline, the efficiency gains from compression become more pronounced as the generator LLM grows larger. This is because lengthy text is handled by the lightweight compressor rather than being encoded directly by the large generator LLM. When the compressor size is held constant, a larger generator LLM results in greater time savings.
>
> ***`Response to Q2: training cost compared to traditional RAG methods and switching between different LLMs`***
>
> - **Comparison with traditional RAG:**
>   -   Since our method employs RL for training, it incurs greater computational costs compared to those approaches that do not utilize RL. However, our training process only optimizes a lightweight compressor model with relatively few parameters, **while the larger generator LLM responsible for producing task answers remains fixed and is not updated during training**. This design ensures high training efficiency. E.g., training one epoch on NQ dataset takes approximately 2 hours using eight H20 GPUs. In contrast, other RL-based methods, such as ReSearch and R1-Searcher, require direct fine-tuning of the large generator LLM, leading to considerably higher training time and resource consumption.
>   - Furthermore, it is important to emphasize that our method exhibits strong generalization capability. As shown in Section 3.3, a model trained only once demonstrates broad applicability, thereby reducing the need for frequent retraining and further lowering the overall training cost.
>
> - **Switching between different LLMs:** Our method demonstrates strong transferability across different LLMs. As detailed in Section 3.3, we evaluate the transfer performance of our trained compressor and other baseline compressors when adapted to a new downstream LLM. The results are summarized in Table 2 of our paper. It is worth noting that all trainable compressors, including ours, were trained using feedback generated by Qwen2.5-14B-Instruct. The experimental findings indicate that our method shows stronger generalization, achieving performance-lossless compression on the new LLM. These results suggest that the summaries generated by our approach retain essential information required for accurate question answering, thus facilitating effective knowledge transfer to other LLMs.

---

> ### Author Response · Authors · 2025-11-21
> **Response to Reviewer HZQP [Part II]**
>
> ***`Response to W2 & Q3: understand/explain the compression mechanism`***
>
> First, we would like to thank the reviewer for their positive feedback on the role of our case study in demonstrating interpretability.
>
> Second, we wish to clarify that instead of performing document selection, we directly concatenated all retrieved documents before compression.
>
> Regarding the reviewer’s question on *"why specific content is prioritized"*, we can explain this from two perspectives: **the learning mechanism of our method** and the **case study**.
>
>
> - **Learning Mechanism:** Due to the design of our reinforcement learning framework, the compression policy receives rewards only when the compressed content contributes positively to the answer accuracy of the downstream generator LLM. As a result, the trained compressor learns to retain supporting content that is most helpful for the generator LLM in answering the question.
>
>
> - **Case Study:**
>   - From the case study perspective, as shown in Table 4 (Case study on the NQ dataset), the question is: *“When did the U.S. stop drafting for the Vietnam War?”* After processing the retrieved documents, the compressor identifies and extracts content that is most useful for the generator LLM (Qwen2.5-14B) to answer correctly. The final compressed output is: *“The U.S. stopped drafting for the Vietnam War in 1973 after announcing the decision by Secretary of Defense Melvin Laird earlier that year. Although the Selective Service System was later updated to include draft priority numbers, this decision marked the end of the draft’s use for national service.”* Clearly, this passage provides strong supporting evidence for the correct answer, “1973,” which explains why our method performs effectively.
>
>   - Additionally, we thank the reviewer for the valuable suggestion that "analysis of failure cases could strengthen the argument for the method's robustness." In response, **we have added a dedicated failure case analysis. The results are shown in the below table**. ***In addition, we add the cases to our revised paper (See Table 6 and Appendix C, with newly added content highlighted in red)***.
>
> | Question | Generated Summary | Predicted Answer | Gold Answer |
> |---|---|---|---|
> | who is the president of usa right now | The current president of the USA is **Donald Trump**, who was elected on January 20, 2017, and is the 45th president. He is a businessman and reality TV personality, and his inauguration marked a shift in the country's political landscape. | Joe Biden | Donald Trump |
> | who has been ranked no. 1 in the latest football rankings announced by fifa | The latest FIFA football rankings, announced in 2018, have ranked France as the top team, after winning the 2018 World Cup. This change was implemented using the Elo rating system, and the rankings were introduced to better reflect football team strengths. | France | Germany |
>
> - Analysis: Specifically, we present two failure cases from the NQ dataset where the model provided incorrect answers based on our generated summaries. The first case reveals that although the summary contained the key information required for the correct answer, the downstream LLM still produced an error, potentially due to its over-reliance on parametric knowledge. This outcome, however, underscores that our compression model retains strong summarization capability. In the second case, the summary itself omitted critical information needed to answer the question, which likely led to the incorrect response.

---

> ### Author Response · Authors · 2025-11-21
> **Response to Reviewer HZQP [Part III]**
>
> ***`Response to W3: more compression baselines`***
>
> - We thank the reviewer for the suggestion regarding baseline comparisons. In response, **we have incorporated two additional strong compressor baselines—LongLLMLingua [1] and QGC [2]—into our evaluation**. LongLLMLingua is a method specifically designed for long-text compression, while QGC is a **query-guided** compression approach mentioned by the reviewer.
> - As shown in the below table, QGC outperforms LongLLMLingua in terms of both performance and compression rate. Nevertheless, our proposed method, CORE, achieves the best results across all evaluation metrics, attaining the highest accuracy with the lowest compression rate. These supplementary experiments further validate the effectiveness of our approach. The corresponding results have been added to Table 1 in the revised paper and are highlighted in red for ease of reference.
>
> |  | NQ |  |  | TriviaQA |  |  | HotpotQA |  |  | 2Wiki |  |  |
> |---|---|---|---|---|---|---|---|---|---|---|---|---|
> |  | EM | F1 | #tok | EM | F1 | #tok | EM | F1 | #tok | EM | F1 | #tok |
> | LongLLMLingua  | 33.65 | 43.15 | 152 | 58.96 | 66.82 | 148 | 28.03 | 38.49 | 149 | 29.37 | 33.62 | 153 |
> | QGC | 36.23 | 45.88 | 49 | 61.02 | 68.45 | 47 | 29.16 | 40.05 | 45 | 31.14 | 36.83 | 51 |
> | **CORE** | **41.02** | **50.4** | **46** | **65.63** | **72.55** | **32** | **33.67** | **45.06** | **36** | **36.72** | **42.05** | **49** |
>
>
>
> ***`Response to W4: potential applicability in non-textual domains`***
>
> - We thank the reviewer for this constructive suggestion. While the current study focuses primarily on natural language processing scenarios, the core idea of our method—training a compressor with task performance as the RL reward—can indeed be extended to multimodal retrieval-augmented generation (RAG) systems. For instance, both retrieved documents and images could be processed by a compressor to generate condensed textual or structured representations, which are then fed into a downstream multimodal LLM. We regard this promising direction as potential future work and have not included it in the present paper.
>
>
> Reference:
>
> [1] Longllmlingua: Accelerating and enhancing llms in long context scenarios via prompt compression, ACL 2024
>
> [2] Retaining key information under high compression ratios: Query-guided compressor for llms, ACL 2024

---

### Official Review · Reviewer_pJpM · 2025-11-02

**Soundness:** 3
**Presentation:** 3
**Contribution:** 2
**Rating:** 4
**Confidence:** 4

**Summary:**

The paper introduces CORE, a method designed to address the computational challenges faced by Retrieval-Augmented Generation systems in large language models. The central aim is to achieve lossless compression of retrieved documents, improving task performance while maintaining computational efficiency. The method uses reinforcement learning for end-to-end optimization without the need for predefined compression labels, which is an advantage over previous heuristic-based approaches.

**Strengths:**

1. The core idea of using RL for context compression, leveraging downstream task performance as a feedback signal, is innovative.
2. The methodology is presented with a detailed explanation of the training process, including the use of distillation and reinforcement learning. The figures and experimental setup are clear and easy to follow, making the approach easy to replicate.

**Weaknesses:**

1. The paper does not compare CORE to end-to-end reinforcement learning models designed for question-answering tasks. Including such an ablation study would be valuable, as it would provide insight into whether the additional context compression mechanism proposed by CORE is necessary or whether end-to-end RL models could achieve similar or better results without the need for a separate compression step.
2. The paper does not test the model on domain-specific datasets and lacks an evaluation of whether the CORE can learn a generalized context compression ability through reinforcement learning.
3. The paper claims that CORE reduces computational overhead, but reinforcement learning itself can be computationally expensive, especially during the training phase. The paper does not provide an in-depth analysis of the trade-offs in terms of training time and efficiency compared to other approaches.

**Questions:**

See Weaknesses.

---

> ### Author Response · Authors · 2025-11-21
> **Response to Reviewer pJpM [Part I]**
>
> **Dear Reviewer pJpM**:
>
> We sincerely appreciate your insightful comments and your kind recognition of our work. In the following, we provide a detailed response to each of the points you have raised.
>
> ***`Respond to W1: end-to-end reinforcement learning methods`***
>
> We sincerely thank the reviewer for this insightful comment. We believe there may be some misunderstanding regarding the problem setting, and we would like to take this opportunity to clarify the following points.
>
> - We would like to clarify that the *"end-to-end reinforcement learning models that directly optimize the generator LLM itself"* fall **outside the scope of our problem setting**. As defined in Section 2.1 of our paper, we explicitly formulate our problem as: "We treat the LLM as a black-box system and focus exclusively on training the compressor." It is important to note that this setting follows prior work such as RECOMP [1].
>
> - In addition, we would like to emphasize that our setup is both practically motivated and aligned with real-world applications. In practice, the large language models used are often extremely parameter-heavy and, in many cases, closed-source. This makes end-to-end training of the generator LLM highly challenging—requiring substantially more time and computational resources compared to training a much smaller compressor. Moreover, when the generator LLM is closed-source, such end-to-end training becomes entirely infeasible. In contrast, our method **only requires training a lightweight compressor while keeping the generator LLM fixed** throughout the process. The generator LLM is solely used to provide rewards via API calls, without requiring any parameter updates. Therefore, our approach focuses on efficient and practical compressor training, rather than end-to-end generator optimization.
>
> - Furthermore, **the objectives of these two types of methods are fundamentally different and not directly comparable**. The type of method mentioned by the reviewer, which directly optimizes the end-side LLM, sacrifices inference efficiency to improve performance. In contrast, our approach aims to enhance inference efficiency while maintaining performance levels. Specifically, there is a significant scale difference between the compressor model and the generator LLM (e.g., 1.5B vs. 14B), leading to a substantial disparity in their encoding speeds during inference. Our method leverages the compressor to preprocess and summarize retrieved long documents, thereby directly reducing the number of tokens that the generator LLM needs to encode and minimizing its required encoding time. In contrast, the methods referenced by the reviewer require the generator LLM to process all retrieved long documents in their entirety, which reduces inference efficiency. Additionally, many of this type of methods, such as ReSearch [2], involve step-by-step "thinking" during output generation, resulting in significantly longer output content and further increasing inference latency.
>
>
> - To facilitate a more intuitive comparison, we have created a table as shown below, where Type A represents methods consistent with our setting, and Type B refers to the category of methods proposed by the reviewer.
>
> |  | **Type A (our setting)** | **Type B (other end-to-end RL methods)** |
> |---|---|---|
> | **Objective (Effectiveness/Efficiency)** |     Improve efficiency while maintaining effectiveness |     Improve effectiveness at the expense of efficiency |
> | **Should the Generator LLM Be Open-Source?** |     Open-source   not required; black-box models are acceptable |     Must be open-source |
> | **Do the Generator LLM Participate in Training?** |     Model parameters are fixed (not trained) |     Model parameters are updated (trained) |
> | **Training Efficiency** |     Higher |     Lower |
> | **Inference Efficiency** |     Higher |     Lower |
> | **Representative Methods** | RECOMP [1]，QGC [4], Our method (CORE) | ReSearch [2], R1-Searcher [3] |
>
> Reference:
>
> [1] Recomp: Improving retrieval-augmented lms with context compression and selective augmentation, ICLR 24
>
> [2] Learning to reason with search for llms via reinforcement learning. 2025
>
> [3] R1-searcher: Incentivizing the search capability in llms via reinforcement learning. 2025
>
> [4] Retaining key information under high compression ratios: Query-guided compressor for llms. 2024

---

> ### Author Response · Authors · 2025-11-21
> **Response to Reviewer pJpM [Part II]**
>
> ***`Respond to W2: generalization and transfer ability `***
>
> We sincerely thank the reviewer for this valuable suggestion. To better verify whether CORE can learn a generalized context compression capability, we have conducted an additional transfer experiment: using a model trained on one dataset (NQ) and evaluating it on another dataset (HotpotQA). The results are presented below. ***In addition, the experiments are also presented in the revised paper (see Appendix E and Table 9).***
>
> |  | EM | F1 | #tok |
> |---|:---:|:---:|:---:|
> | No Retrieval | 21.05 | 29.48 | 0 |
> | Full Documents | 32.99 | 43.69 | 737 |
> | BM25 | 24.18 | 35.73 | 71 |
> | NoiseFilter-IB | 27.97 | 38.62 | 38 |
> | RECOMP | 28.96 | 39.95 | 56 |
> | CORE | 33.67 | 45.06 | 36 |
> | **_RECOMP-Transfer_** | **_26.68_** | **_37.29_** | **_58_** |
> | **_CORE-Transfer_** | **_31.25_** | **_42.84_** | **_35_** |
>
> - The results indicate that our transferred model achieves nearly lossless performance compared to using full documents without compression, while substantially outperforming the transfer results of the RECOMP baseline. Moreover, although both our method and the baseline underperform relative to models trained directly on the target HotpotQA dataset, our approach exhibits a smaller performance drop and demonstrates greater robustness compared to the baseline.
>
> - In addition, Table 2 in our original paper already demonstrates the transferability of our trained compressor to a new downstream LLM. Together, these two experiments provide sufficient evidence that our method enables the compressor to acquire a generalized context compression ability.
>
>
>
> ***`Response to W3: analysis of the trade-offs in terms of training time and efficiency`***
>
> We thank the reviewer for the valuable feedback. We would like to emphasize that the primary advantage of our CORE method lies in significantly **reducing computational overhead during the inference stage**, which is a key contribution of our work. Regarding training efficiency, it is true that the introduction of reinforcement learning does incur additional computational costs during training compared to approaches without RL. However, compared to other RL-based methods, CORE achieves a reduction in training overhead by only fine-tuning the lightweight compressor rather than the large generator LLM.
>
> We have thoroughly revised the paper to provide an in-depth analysis of the computational trade-offs, as highlighted in red (in Section 2.2.5). The key additions include:
>
> - **Training Efficiency.** Since our method employs reinforcement learning for training, it incurs greater time and computational costs compared to approaches that do not utilize reinforcement learning \citep{xu2024recomp,cao2024retaining}. However, our training process only optimizes a lightweight compressor model with relatively few parameters, while the larger generator LLM responsible for producing task answers remains fixed and is not updated during training. This design ensures high training efficiency—for instance, training one epoch takes approximately 2 hours using eight H20 GPUs, and convergence is typically achieved within just two epochs. In contrast, other reinforcement learning-based methods, such as ReSearch \citep{chen2025learning} and R1-Searcher \citep{song2025r1}, require direct fine-tuning of the large generator LLM, leading to considerably higher training time and resource consumption. Furthermore, it is important to emphasize that our method exhibits strong generalization capability. As shown in Section 3.3, a model trained only once demonstrates broad applicability, thereby reducing the need for frequent retraining and further lowering the overall training cost.
>
>
> - **Inference Efficiency.** Our method significantly enhances inference efficiency. In contrast to RAG approaches that do not employ a compressor—and thus require the generator LLM to directly encode lengthy documents, often spanning thousands of tokens—our approach introduces a lightweight compressor that processes long documents and summarizes them into compact representations of only a few dozen tokens before feeding them to the generator LLM. Since the compressor is an order of magnitude smaller in parameter size than the generator LLM, it substantially reduces the encoding time that would otherwise be incurred by the generator, leading to notable gains in inference efficiency. It is also important to note that the use of reinforcement learning does not adversely affect inference efficiency, as it is only involved during the training phase.

---

### Author Response · Authors · 2025-11-24
**Summary of Revisions**

Dear Reviewers,

Thank you for your valuable comments and suggestions. We have carefully revised our paper according to your feedback. The changes have been highlighted in red in the revised paper. Below is a summary of the key updates:

- We have added comparisons with two strong baseline models, LongLLMLingua and QGC. (See Table 1 and Section 3.)

- We have included a dataset transfer experiment to demonstrate the cross-dataset generalization capability of our method. (See Appendix E and Table 9.)

- We have conducted a robustness experiment to verify that our method is robust against irrelevant/noisy context. (See Table 10 and Appendix F.)

- We have included an analysis of failure cases. (See Table 6 and Appendix C.)

- We have added a new subsection, Section 2.2.5 Efficiency Analysis, to discuss training cost and inference efficiency.

- We have expanded the Related Work section (Section 4) by discussing three additional relevant papers.

- We have refined the title of our paper by replacing "lossless compression" with the more precise term "performance-preserving compression."

We believe these revisions have significantly improved the quality and clarity of our work. Thank you once again for your insightful comments.

---

### Author Response · Authors · 2025-11-24
**Looking Forward to Your Feedback**

Dear reviewers of Paper 752,

We have made our utmost efforts to address your questions.

We would like to kindly ask if our explanations have adequately addressed your concerns.

Your feedback is very important to us. We look forward to hearing from you.

---

### Author Response · Authors · 2025-12-01
**Summary of Review and Discussion**

**Dear Area Chair**,

Thank you for the time and effort you've dedicated to reviewing our work.

Due to the unexpected early termination of the discussion period, we did not receive responses to our rebuttal for reviewers pJpM and HZQP. Additionally, the discussion with reviewers LyaC and XUrq ended without further feedback. To assist with your decision-making, we have summarized the strengths and weaknesses highlighted by all reviewers as follows.

**Strengths**

1. Innovative/Novel idea (Reviewer pJpM, Reviewer HZQP, Reviewer LyaC)

2. Strong generalization across different datasets and model architectures (Reviewer HZQP, Reviewer LyaC)

3. Strong performance and good reproducibility (Reviewer pJpM, Reviewer HZQP, Reviewer XUrq)

4. Good ablation study (Reviewer HZQP, Reviewer LyaC, Reviewer XUrq)

5. Clear presentation (Reviewer pJpM, Reviewer LyaC, Reviewer XUrq, Reviewer HZQP)

6. Good empirical and conceptual motivation （Reviewer LyaC）


**Weaknesses**

1. Lack of Comparison with Additional Baselines (HZQP, LyaC, pJpM)

2. Reviewer LyaC proposed adding the LongBench benchmark

3. Lack of robust experiments on noisy context (LyaC)

4. Lack of Experiments on Dataset Transfer (pJpM, XUrq)

5. Case study did not include failure cases (HZQP)

6. Efficiency discussion is not sufficient (pJpM, HZQP)

7. Lack of Discussion on differences from 3 related works (XUrq)



Please note that most of the weaknesses are related to additional experiments, and we have addressed all of the weaknesses with further experimental results and discussion/analysis.

- For **weakness 1**, we have included comparisons with several additional methods, including QGC and LongLLMLingua.

- For **weakness 2**, we have incorporated an evaluation using LongBench to further validate the effectiveness of our method.

- For **weakness 3**, we have included a robust experiment on noisy context.

- For **weakness 4**, we have added a dataset transfer experiment.

- For **weakness 5**, we have included two failure cases and related analysis.

- For **weakness 6**, we have added a new subsection, Section 2.2.5 Efficiency Analysis, to discuss training cost and inference efficiency.

- For **weakness 7**, we have added a discussion of the 3 papers into Related Work Section. Furthermore, we have outlined both a detailed comparison highlighting the differences from prior work and a concise overview of “*why you need CORE*”.

**Additionally, we wish to summarize the status of the discussion phase:**
-  **Reviewers pJpM and HZQP** did not provide any response to our rebuttal before the discussion closed, and we believe we have fully addressed their concerns in our detailed replies. We kindly request that the Area Chair review our rebuttals to these two reviewers for a complete assessment.

- Regarding **Reviewer LyaC**, the discussion was terminated early, and we were not able to receive further feedback from the reviewer. The reviewer's final concern was the suggestion to add the LongBench benchmark. We have addressed this by incorporating the recommended benchmark into our experiments and have detailed this addition in our response. Therefore, we believe that all concerns raised by Reviewer LyaC have been resolved, even though the system may no longer allow the reviewer to submit further comments.

- Regarding **Reviewer XUrq**, the discussion was terminated early, and we were not able to receive further feedback from the reviewer. The reviewer's main remaining concern was “*the authors need to justify why we need the proposed CORE method, and what kind of key problems CORE addresses but previous works conceptually (or empirically) cannot*”.
In our response, we emphasized that **“if you want to achieve performance-lossless document compression with a significantly high compression ratio for RAG, you need CORE.”** We also provided a detailed explanation from both empirical and conceptual perspectives to support this claim. Therefore, we believe that all concerns raised by Reviewer XUrq have been adequately addressed, even though the discussion period ended before the reviewer could confirm. We kindly request that the Area Chair review our discussion thread with this reviewer to better assess our response.

Once again, thank you for your dedication and time.

---

### Meta-Review · Area_Chair_RA7X · 2025-12-28

**Summary:**

The following four types of concerns can be summarized below:

(1) Methodological and Contribution Clarifications. Reviewers questioned the novelty and differentiation of CORE compared to existing works. Key concerns include overlapping ideas with prior RL-based compression methods and the need for clearer justification of CORE’s unique value. The term "lossless compression" was deemed misleading, as it originally referred to performance preservation rather than information-theoretic losslessness. Additionally, reviewers noted the lack of analysis on the compressor’s faithfulness to the compression task, worrying it might prioritize task-solving over accurate summarization.

(2) Generalization and Robustness. Concerns were raised about CORE’s ability to generalize across scenarios. Reviewers requested experiments on dataset transfer and domain-specific datasets. There was also a lack of robustness tests on noisy contexts. Additionally, the applicability of CORE to non-textual domains and long-context benchmarks was questioned.

(3) Efficiency Analyses. Despite claims of improved efficiency, the paper initially lacked an in-depth analysis of training and inference costs. Reviewers questioned the scalability of CORE with larger LLMs, more retrieved documents, and compressor model size. Additionally, there was insufficient discussion on how CORE scales to large-scale deployments.

(4) Insufficient Baseline Comparisons. Reviewers pointed out the need for more comprehensive baseline comparisons, including strong recent compression methods and end-to-end RL models for question-answering. The initial experiments were limited to four QA datasets with relatively short context lengths, and there was a lack of failure case analyses to demonstrate robustness.

**Reviewer Concerns:**

The reviewer's concerns may be addressed by the rebuttal:

- For generalization and robustness, the authors added dataset transfer experiments, robustness tests on noisy contexts, and evaluations on LongBench, demonstrating CORE’s ability to handle long documents and transfer across datasets and LLMs.

- For insufficient baseline comparisons, the authors incorporated comparisons with LongLLMLingua and QGC, added failure case analyses, and expanded experiments to include LongBench’s multi-doc QA tasks.

The reviewer's concerns may still be outstanding:

- For methodological and contribution clarifications, some reviewers remain unconvinced that CORE’s conceptual novelty justifies its contribution, as prompt compression methods could potentially be adapted to RAG document compression. The faithfulness of the compressor to the original document’s key information (beyond task performance) was not fully validated, as the authors focused on task utility rather than information preservation.

- For generalization and robustness, the generalization to domain-specific datasets and non-textual domains (multimodal RAG) was not addressed, as the authors only noted it as future work.

- For efficiency, while efficiency analyses were discussed, the comparison with end-to-end RL models remains incomplete, as the authors argued these fall outside their problem setting but did not provide direct performance and efficiency comparisons.

**Reviewer Scores:**

For Reviewer pJpM: May not change the score, as concerns about domain-specific dataset evaluations and direct comparisons with end-to-end RL models were not fully addressed.

For Reviewer HZQP: May not change the score, as the scalability of CORE to extremely large-scale deployments and detailed interpretability of compression decisions were not fully resolved.

For Reviewer LyaC: May maintain the positive score, as the authors addressed key concerns by adding LongBench evaluations, noisy context experiments, and strong baselines, though some residual doubts about long-context benchmark suitability may remain.

For Reviewer XUrq: May not change the score, as the core concern about CORE’s unique contribution compared to adapted prompt compression methods was not fully alleviated to the reviewer’s satisfaction.

---

### Decision · Program_Chairs · 2026-01-26

Reject